# Patchy striatonigral neurons modulate locomotor vigor in response to environmental valence

**Sarah Hawes[1], Bo Liang[2,3], Braden Oldham[1], Breanna T Sullivan[1], Lupeng Wang[1], Bin Song[1], Lisa Chang[1], Da-Ting Lin[2,4]\*, Huaibin Cai[1]\***

[1]Transgenic Section, Laboratory of Neurogenetics, National Institute on Aging, National Institutes of Health, Bethesda, United States; [2]Intramural Research Program, National Institute on Drug Abuse, National Institutes of Health, Baltimore, United States; [3]School of Electrical Engineering & Computer Science, College of Engineering & Mines, University of North Dakota, Grand Forks, United States; [4]The Solomon H. Snyder Department of Neuroscience, Johns Hopkins University School of Medicine, Baltimore, United States

**\*For correspondence:**
da-ting.lin@nih.gov (D-TL);
caih@mail.nih.gov (HC)

**Competing interest:** The authors declare that no competing interests exist.

## eLife Assessment

The manuscript provides **important** findings on how striatal projection neurons regulate spontaneous locomotion speed in the context of implicit motivation and distinct contextual valence. The manuscript presented **convincing** supporting evidence for the findings. This work will be of broad interest to neuroscientists in the fields of basal ganglia, movement control, and cognition.

**Abstract** Spiny projection neurons (SPNs) in the dorsal striatum play crucial roles in locomotion control and value-based decision-making. SPNs, which include both direct-pathway striatonigral and indirect-pathway striatopallidal neurons, can be further classified into subtypes based on distinct transcriptomic profiles and cell body distribution patterns. However, how these SPN subtypes regulate spontaneous locomotion in the context of environmental valence remains unclear. Using Sepw1-Cre transgenic mice, which label a specific SPN subtype characterized by a patchy distribution of cell bodies in the dorsal striatum, we found that these patchy striatonigral neurons constrain motor vigor in response to valence differentials. In a modified light/dark box test, mice exhibited differential walking speeds between the light and dark zones. Genetic ablation of these patchy SPNs disrupted restful slowing in the dark zone and increased zone discrimination by speed. In vivo recordings linked the activity of these neurons to zone occupancy, speed, and deceleration, with a specific role in mediating deceleration. Furthermore, chemogenetic activation of patchy SPNs—and optical activation of striatonigral neurons in particular—reduced locomotion and attenuated speed-based zone discrimination. These findings reveal that a subtype of patchy striatonigral neurons regulates implicit walking speed selection based on innate valence differentials.

## Introduction

Motor control relies on the complementary and simultaneous regulation of downstream nuclei by the striatal direct and indirect pathways (*Kravitz et al., 2010*; *Cui et al., 2013*; *Barbera et al., 2016*; *Yttri and Dudman, 2016*). Striatal projection neurons (SPNs) are further hypothesized to encode implicit motivation through the regulation of action vigor, such as movement speed (*Dudman and*

*Krakauer, 2016*). However, the mechanisms by which striatal microcircuitry achieves this remain poorly understood.

A subpopulation of SPNs, referred to as 'patch' or 'striosome' neurons, is organized into distinct patchy clusters of varying sizes within the broader striatal matrix (*Crittenden and Graybiel, 2011*; *Gerfen and Surmeier, 2011*). Both patch and matrix compartments comprise a mixture of direct and indirect pathway SPNs (*Crittenden and Graybiel, 2011*; *Gerfen and Surmeier, 2011*). Despite this similarity, differences in neurochemical and genetic composition, along with distinct interactions with surrounding circuitry, suggest that patch and matrix territories have specialized roles in regulating behavior (*Graybiel and Matsushima, 2020*). Recent behavioral studies highlight the unique roles of patchy SPNs, implicating them in value-based learning and risk assessment (*Friedman et al., 2015*; *Friedman et al., 2020*). However, the interplay between these valuative functions and the conventional role of the striatum in locomotor control remains poorly understood. Given their distinctive characteristics, patchy SPNs are a compelling candidate for linking external valence to implicit motivational processes, such as action vigor and movement speed.

Studies of patchy SPNs make it clear these neurons are themselves diverse, and that accessing this population in varied ways necessarily results in the study of various subpopulations. Patchy SPNs were first distinguished from the surrounding matrix based on gradients in histochemical markers (*Pert et al., 1976*; *Gerfen, 1984*; *Graybiel, 1984*). They have since been accessed in vivo by relying on prenatal injections, by targeting patchy SPN-preferring afferents, and most recently through the development of genetically modified mouse lines with transgene expression controlled by genes concentrated in patchy SPNs (*Brimblecombe and Cragg, 2017*; *Prager and Plotkin, 2019*). For the present study, we selected the Sepw1-Cre (NP67) BAC transgenic line to label the patchy SPNs (*Gerfen et al., 2013*). These patchy SPNs include both direct (80%) and indirect (20%) pathway neurons and, unlike the matrix, receive innervation by the anxiety-modulating bed nucleus of the stria terminalis (*Smith et al., 2016*). This line is well characterized to express Cre in roughly 15% of striatal territory, representing similar striatal coverage to that derived from classic histological characterization of patchy SPNs (*Smith et al., 2016*).

Using Sepw1-Cre mice, we tested whether the assessment or response to naturalistic contrasts in valence depends on patchy SPNs. We employed a modified Light/Dark box paradigm combined with genetically targeted ablation and enhancement, in vivo imaging of patchy SPN somas and terminals, and synapse-specific optogenetics. Our findings reveal that patchy SPNs control the speed at which mice navigate the valence differential between high- and low-anxiety zones, without affecting valence perception itself. This provides the first evidence that patchy SPNs regulate naturalistic motor behavior and demonstrate that they drive implicit motivation to align situational valence with speed selection.

## Results

### Genetic ablation of patchy SPNs in Sepw1-Cre mice

To investigate the impact of patchy SPNs on behavior, we employed Sepw1-Cre (NP67) BAC transgenic mice (*Gerfen et al., 2013*; *Smith et al., 2016*) and bilaterally injected either adeno-associated viral vector, AAV-DIO-taCasp3, to create dorsal striatal patch ablation (PA mice), or AAV-DIO-mCherry (controls) (*Figure 1A*). Cre-dependent mCherry signals were colocalized with the patch marker μ-opioid receptor (MOR1) in the dorsal striatum of controls (*Figure 1—figure supplement 1*). Examination of MOR1-positive territories following behavior revealed an approximately 60% loss of patchy neurons in the dorsal striatum of PA mice compared to control mice (p<0.0001, *Figure 1B–C*). By contrast, no loss of MOR1-positive territories was observed in the ventral striatum of either PA or control mice (*Figure 1B–C*).

### Light/dark box valence differential impacts speed

The Light/dark box (LDb) test relies on the natural preference of nocturnal mice for a sheltered, dark zone despite also exploring a more anxiogenic, light zone (*Bourin and Hascoët, 2003*). We differentially sub-lit a U-shaped maze to mitigate light exchange across zones (*Figure 1D*), while permitting uninterrupted areal video recording. Behavior in control mice verified dark preference, as demonstrated by greater stay time in the dark zone. Similar to controls, PA mice also demonstrated

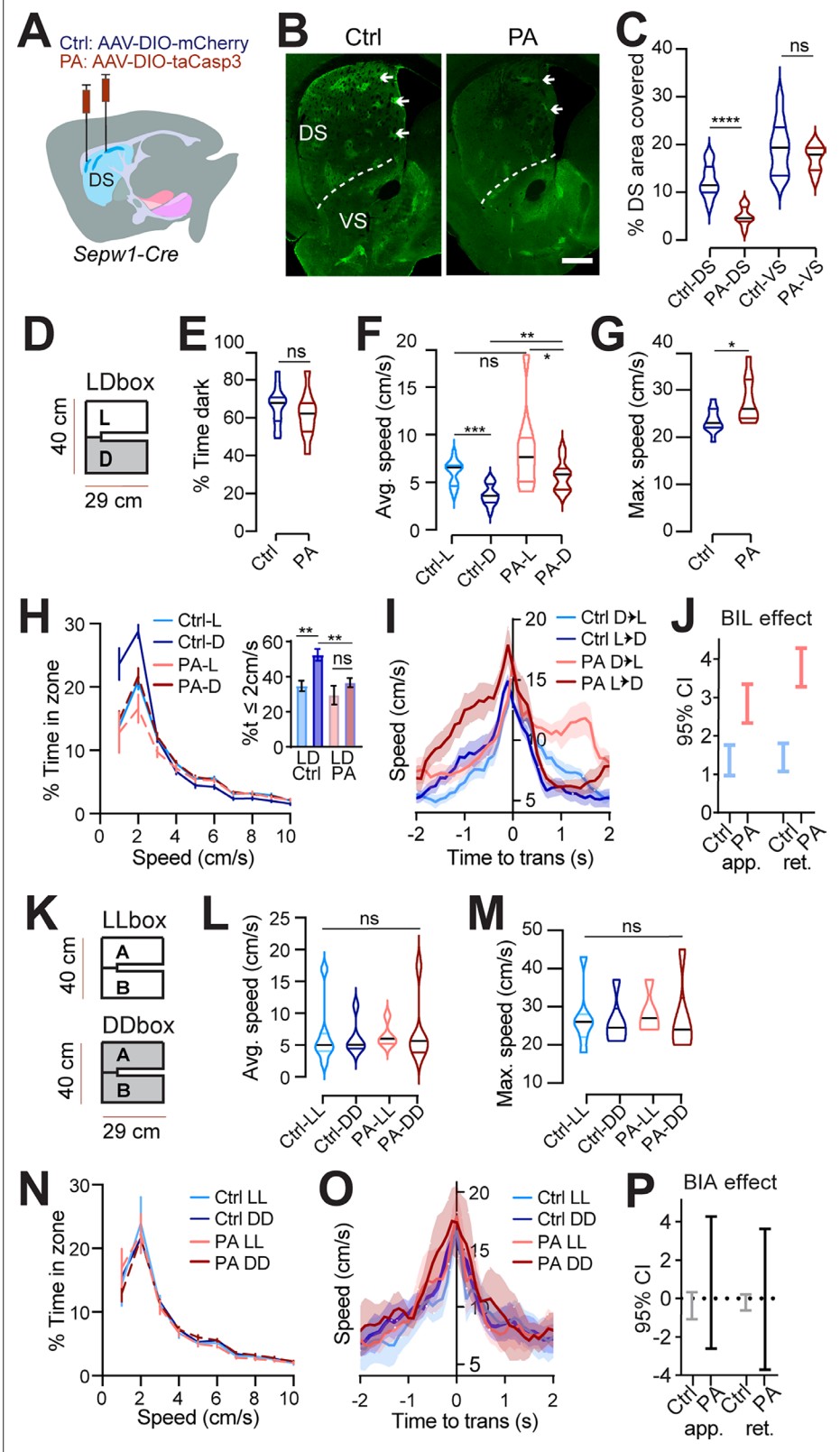

**Figure 1.** Patchy spiny projection neuron (SPN) ablation reduces rest and unmasks anxious vigor at choice points. (**A**) Injection schematic. (**B**) MOR1 staining. Scale bar: 500 µm. (**C**) MOR1 quantification to assess ablation. 3 mice per group, n=22 Control (Ctrl) and 30 patchy SPN ablated (PA) hemisections. For dorsal striatum (DS), PA vs Ctrl, two-tailed Mann-Whitney, ****p<0.0001. (**D–J**) Light/Dark box. n=11 (7Male [M],4Female[F]) Ctrl and 10 (6M,4F) PA

*Figure 1 continued on next page*

*Figure 1 continued*

mice. (**D**) LDbox schematic. (**E**) % time in dark. (**F**) Average speed. Following RM ANOVA, post hoc comparisons, ***p=0.001, *p=0.0371; **p=0.0015. (**G**) Maximum speed. Two-tailed Mann-Whitney, *p=0.0434. (**H**) Speed distribution normalized to zone. Insert, %time ≤2 cm/s, paired t-test, L vs D Ctrl **p=0.001, PA p>0.05; unpaired t-test, Ctrl. vs. PA in dark, **p=0.0018, in light p>0.05. (**I**) Transition speed. (**J**) Generalized Linear Mixed-Effect (GLME) 95% C.I. for impact of body-in-light (BIL) on transition speed. (**K–P**) LL/DD box. n=7 (4M,3F) Ctrl-LL, 6 (3M,3F) Ctrl-DD, 5 (3M,2F) PA-LL, 6 (3M,3F) PA-DD mice. (**K**) LL/DDbox schematics. (**L**) Average speed of LL/DDbox test. Mixed Effects analysis, ns. (**M**) Maximum speed of LL/DDbox test. Mixed Effects analysis, ns. (**N**) Speed distribution normalized to side 'A' of LL or DD box. (**O**) Transition speed. (**P**) GLME 95% CI for impact of BIL on transition speed overlaps zero for both groups.

The online version of this article includes the following figure supplement(s) for figure 1:

**Figure supplement 1.** Patchy distribution of mCherry-positive cells in the dorsal striatum of Sepw1-Cre mice.

dark preference through greater stay time in the dark zone (*Figure 1E*). Speed was higher during exploration of the anxiogenic light zone for both control and PA mice, despite higher dark zone speed following ablation (L vs D control p=0.001, PA p=0.0371; control vs PA in dark p=0.0015, *Figure 1F*). The similarity of PA mice to controls in terms of dark preference and relatively greater light zone speed indicates that patch SPNs are not critical for basic discrimination of situational valence.

## Ablation of patchy SPNs reduces rest and unmasks anxious vigor at choice points

While zone preference was normal, average speed in the dark zones and the maximum speed obtained were higher for PA mice than for control mice (dark zone speed p=0.0015, max speed p=0.0434, *Figure 1F–G*). Importantly, the reflection of valence in speed distributions between light and dark zones, present in controls, was lost for PA mice. The loss was due to a significant reduction in the percent time spent at restful speeds (≤2 cm/s) in the dark zones among PA mice relative to controls (p=0.0018, *Figure 1H*). Given that PA mice retained zone discrimination and dark preference, reduced stillness in the dark zones suggests that ablation of patchy SPNs leads to a failure to slow down or rest under safe conditions.

Zone transitions were accompanied by speed peaks for all mice (*Figure 1I*), suggesting invigorated action selection at this choice point. For controls, differential speed due to surrounding valence is immediately apparent in this moment of transition, with greater speeds expressed on either side of a transition accompanying the body-in-light (BIL, i.e. preceding a transition into dark and following a transition into light) versus body-in-dark (BID) moments. As a proxy for valence differential, we defined BIL effect as the difference between BIL and BID at transition moments (See Methods). We then used a Generalized Linear Mixed-Effects (GLME) model to estimate the effects of different factors on speed during these transitions. Both factors of BIL and PA mice were shown to increase speed at this choice point (GLME coefficient fixed effects 95% CI for speed increase by PA [0.73218, 3.9241] and BIL [1.7464, 2.3911] on approach, or by PA [2.2344, 2.8761], and BIL [0.89109, 3.593] on retreat). We dug deeper into the BIL effect on transition speed over the two groups of mice separately. As with controls, PA mice responded to a shift in valence accompanying zone transitions with acutely elevated speed during BIL windows. However, BIL had a larger effect on transition speed for PA mice than for controls, indicating that PA mice exhibited elevated transition discrimination through velocity (GLME fixed effects 95% CI for impact of BIL on speed: PA approach [2.3376 3.347], retreat [3.2809, 4.2788]; control approach [0.9698, 1.7612], retreat [1.0784, 1.8056], *Figure 1J*). Collectively, the pattern of excess speed in PA mice suggests that patchy SPNs naturally lower speed, supporting rest in the dark and alleviating anxious vigor surrounding transitions.

## Speed modulation by patchy SPNs depends on a valence differential

LDb data implicate patchy SPNs in lowering speed without determining if speed reduction depends fundamentally on light level or on differential light levels and the associated choice in situational valence. To distinguish these possibilities, we tested mice in chambers having the same physical dimensions as the LDb but with homogeneous illumination (*Figure 1K*). Half the cohort was tested in a uniformly brightly lit 'LLb' chamber and the other half in a uniformly dim 'DDb' chamber, with all mice receiving the alternative test one to two weeks later. We found no substantial difference between

LLb from DDb performance in terms of average or maximum speed for either group (*Figure 1L–M*). Moreover, no difference in speed distribution between LLb and DDb was apparent (*Figure 1N*). Local speed peaks persisted while transitioning between equally illuminated zones, perhaps due to the decision to turn a corner or pass through a relatively constricted point joining larger rooms. Yet these transition speeds did not differ between LLb and DDb (*Figure 1O–P*, 'BIA' denotes body in start side 'A,' analogous to BIL as mice are started in LDb light). Thus, the zone-specific LDb speed profile is not related simply to light level but is dependent on a light versus dark valence differential. Therefore, the LLb/DDb data help to establish LDb speed as a behavioral reflection of differential situational valence. These data also indicate that PA elevates valence-specific speed without inducing general hyperactivity. These data support the utility of the modified LDb to test implicit speed choice during valence-driven locomotor performance. Collectively, these data show that locomotor speed modulation by Sepw1+ patchy neurons depends on the presence of a salient valence differential.

## Dark preference and greater light-zone speed are preserved during in vivo calcium imaging

A majority of Sepw1-Cre positive patchy SPNs are the direct-pathway neurons known for promoting locomotion (*Smith et al., 2016*). However, we found that ablation of patchy SPNs increased speed during LDb navigation (*Figure 1F and G*), indicating an inhibitory role of patchy SPNs in regulating locomotion. To investigate the relationship between patchy SPN activity, valence, and speed, we analyzed in vivo calcium transients in patchy SPNs using the genetically encoded calcium indicator GCaMP6s (*Chen et al., 2013*) in 10 miniscope-mounted mice navigating the LDb (*Figure 2A and B*). Histology following GCaMP6s transduction and behavior confirmed 1 mm diameter GRIN lens position on the top of GCaMP-positive patchy neurons in the dorsal striatum (*Figure 2C*, *Figure 2—figure supplement 1A–C*). Cells were identified, and calcium traces (ΔF/F) were extracted (*Figure 2C*, see Materials and methods) using CaImAn (*Pnevmatikakis et al., 2016*; *Pnevmatikakis and Giovannucci, 2017*; *Giovannucci et al., 2019*). Aside from a lack of separation in speeds approaching transitions (possibly due to burden of miniscope wearing), the 10 imaged mice demonstrated the LDb speed phenotype consistent with controls in our ablation experiment (*Figure 2D*, *Figure 2—figure supplement 1D-I*).

## Patchy neuron activity reflects zone, speed, and deceleration

We next analyzed the relationship among patchy neuron activity, zone preference, and locomotor speed. For each neuron, zone preference was determined through comparison of calcium activity in light zone to that in dark zone (see Materials and methods). Neurons showing preferential activation in the light zone substantially outnumbered neurons preferentially active in the dark zone (*Figure 2E and F*). A recent study showed that striatal neurons display heterogeneous responses to movement speed, exhibiting both linear and nonlinear relationships (*Fobbs et al., 2020*; *Dong et al., 2025*). Similarly, we identified speed-tuned cell subtypes with both linear (positive L+ and negative L-) and quadratic (positive Q+ and negative Q-) speed-tuning profiles (*Figure 2G and H*; Materials and methods). Approximately 53% of neurons demonstrated such speed-activity relationships (L+, L-, Q+, and Q-), with the largest fraction of speed-related cells having a quadratic negative relationship (*Figure 2H*). Collectively, 76% of neurons (*Figure 2—figure supplement 1J*) could be defined by these relationships to zone and/or speed, and the distribution of neurons among these categories was highly consistent across mice (*Figure 2I and J*). Light preference and speed relationships coincided within neurons more often than predicted by chance ($\chi^2$ (3, n=532)=78.63, p<0.0001, *Figure 2—figure supplement 1K*). Interestingly, more than half of light-related neurons possess a Q-relationship to speed (*Figure 2K*). These data suggest that many patchy neurons are engaged in speed modulation, particularly within the anxiogenic light.

We next tested for a relationship between calcium activity and change in locomotor speed. To do this, we identified periods of time in which each mouse accelerated or decelerated (see Materials and methods). For each neuron, we measured the activity acceleration preference using receiver operating characteristic (ROC) analysis, similar to that used in previous studies (*Britten et al., 1992*; *Li et al., 2017*; *Fobbs et al., 2020*) (see Materials and methods). For most patchy neurons, we found a strong relationship between deceleration and calcium activity, represented through acceleration/deceleration preference index (ADI) (computed with area under ROC curve,

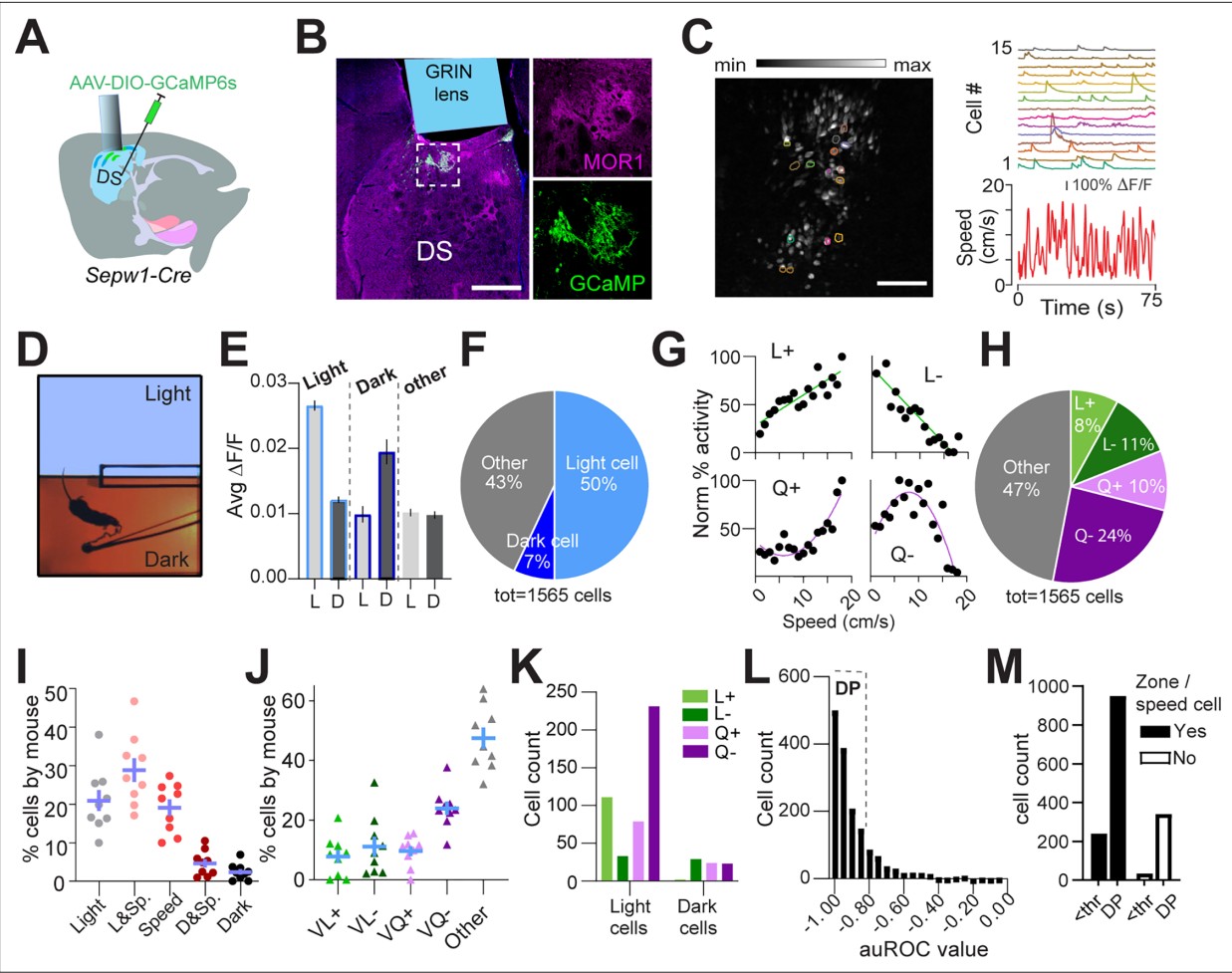

**Figure 2.** Patchy spiny projection neuron (SPN) activity reflects zone, speed, deceleration. Single-cell calcium imaging with miniscopes. n=10 mice (7 M,3F). (**A**) Schematic illustrating placement of GCaMP6s and GRIN lens. (**B**) Example lens placement above imaged patchy SPNs. Selection expanded at right shows GCaMP6s and MOR1 overlap. Scale bar: 500 µm. (**C**) Left: image stack collected through GRIN lens with location of example cells circled. Scale bar: 500 µm. Right: calcium transients from example cells circled at left plotted above speed. (**D**) Light/dark box aerial view with tethered mouse. (**E**) Average activity of zone-preferring (Light or Dark) and non-discriminating (Other) neurons in light 'L' vs dark 'D' zones. Paired t-test p<0.0001 Light cells; p<0.0001 Dark cells; p=0.016 Other cells. (**F**) Most zone-discriminating neurons are light-preferring. (**G**) Sample neurons illustrating speed relationships; clockwise from top left: linear+ (L+), linear- (L-), quadratic+ (Q+), quadratic- (Q-). (**H**) Distribution of speed relationships among imaged neurons. (**I, J**) Distribution is similar across all mice (n=9, excluding one mouse with fewer than 80 cells) for (**I**) zone and/or speed-related neurons (**J**) specific speed relationships. (**K**) Speed relationships among light- and dark-preferring neurons. (**L**) Histogram of acceleration/deceleration preference index (ADI). Strong deceleration-predicting 'deceleration-prediction (DP)' neurons are defined by |ADI|>a threshold 'thr' of 0.8. (**M**) DP neurons are over-represented among zone/speed free neurons, $\chi^2(1,1565)=24.33$, p<0.0001.

The online version of this article includes the following figure supplement(s) for figure 2:

**Figure supplement 1.** MOR1 thresholding, Scope-mounted Light/dark box behavior, and Ca$^{2+}$ imaging supplemental.

see Materials and methods) values close to –1 (*Figure 2L*). We then divided neurons into those with strong deceleration-prediction (DP, |ADI|>0.8, n=1291 neurons) or other neurons (|ADI|<0.8, n=274 neurons). Those neurons for which we had been unable to identify a speed or zone relationship were disproportionately apt to be DP cells ($\chi^2(1,1565)=24.33$, p<0.0001, *Figure 2M*), providing for categorization of additional neurons. As a result, we were able to ascribe a significant relationship to some combination of zone, speed, or deceleration to approximately 98% (1531 of 1565 neurons) of imaged patchy neurons. Importantly, these data corroborate our finding that ablation of patchy neurons increased speed at select moments in the LDbox by showing that many imaged patchy neurons act in concert with locomotor deceleration.

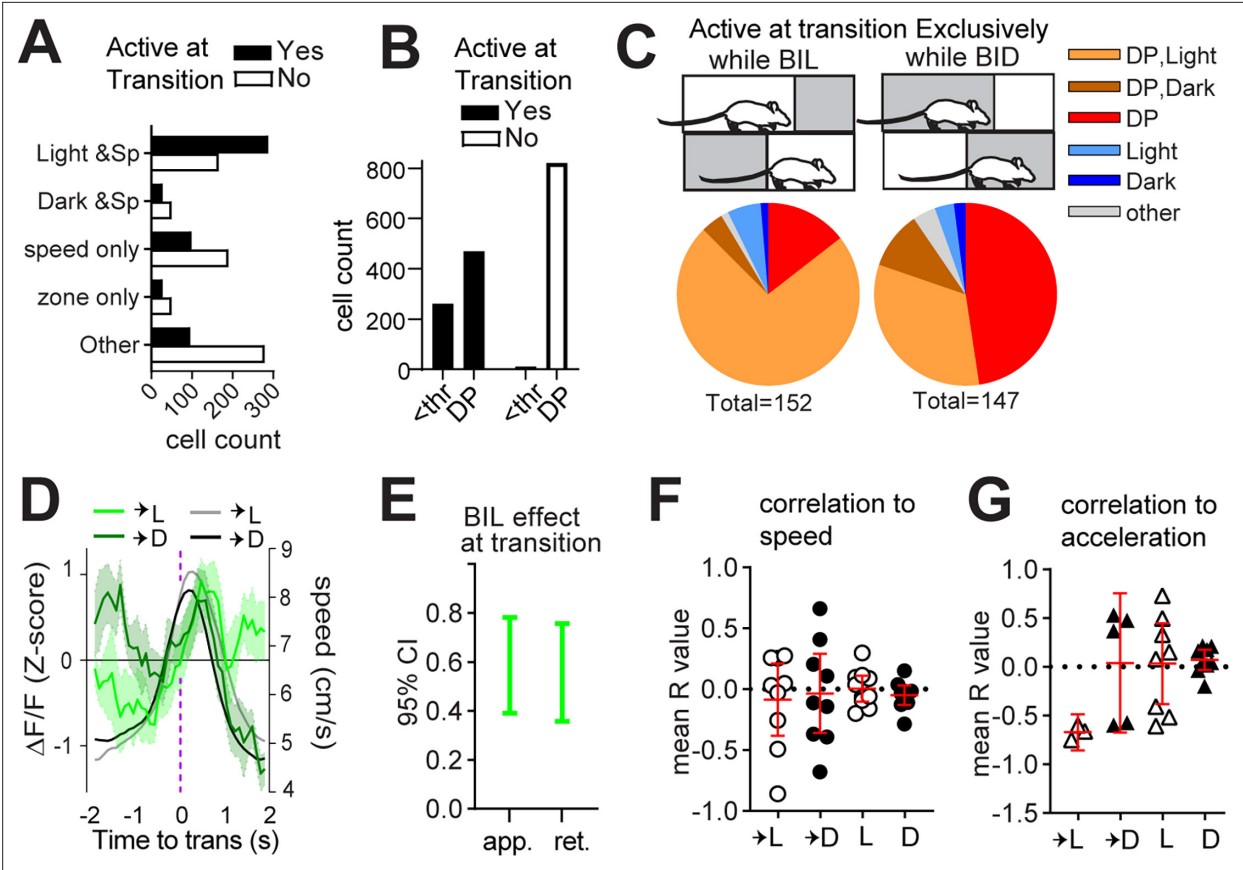

**Figure 3.** Light sensitivity and speed modulation are reflected in transition-active neurons. Single-cell calcium imaging with miniscopes. n=10 (7M,3F) mice. (**A**) A relationship exists between zone/speed encoding and zone transitions, $\chi^2(4, n=1563)=137.3$, $p<0.0001$. (**B**) A relationship exists between deceleration encoding and zone transitions, $\chi^2(1, n=1565)=24.33$, $p<0.0001$. (**C**) Comparison of transition-active neurons selective for either body-in-light (BIL) or body-in-dark (BID). (**D**) Overlay of average $\Delta F/F$ (green, z-score) with mean transition speed (gray scale, cm/s). (**E**) Effect of BIL while approaching or retreating from zone transitions, GLME fixed effects 95% CI (a.u.). (**F, G**) 95% CI shown in red. (**F**) Mean R value per mouse for all acceleration events with significant ($p<0.05$) $\Delta F/F$ cross-correlation to speed (n=9 into light, 9 into dark, 10 within light, 10 within dark), or (**G**) acceleration (n=3 into light, 5 into dark, 8 within light, 9 within dark).

## Light sensitivity and speed modulation are reflected in transition-active neurons

Given that ablation of patchy neurons induced excess locomotor vigor with respect to restful navigation in dark and during BIL transition periods, we hypothesized that patchy neurons active at these time points would possess negative correlations to speed. Limited neuronal activity at predominantly resting states in the dark zone meant that we were unable to perform meaningful analysis of neuron types in these moments, and we focused instead on zone transitions. We found a significant interaction between transition-active status and zone/speed relationships ($\chi^2(4, n=1565)=137.3$, $p<0.0001$, *Figure 3A*) such that cells jointly encoding both light and speed were most abundant among transition-active neurons. Meanwhile, purely speed-related cells, dark and speed-related cells, and cells with no identified zone or speed relationship were most abundant among neurons *not* active at transitions. The most common speed relationship remained quadratic-negative among transition-active neurons, like all neurons (*Figure 2—figure supplement 1J, L*). While deceleration prediction disproportionately characterized neurons not active at transition ($\chi^2(1, n=1565)=24.33$, $p<0.0001$, *Figure 3B*), most transition-active neurons were also DP (*Figure 3B*). Comparison of neurons exclusively active while BIL or else selectively active while BID at transitions illustrated both sensitivity to zone, as well as a predominance of DP among zone-discriminating neurons at the transition moment (*Figure 3C*). This was obscured when considering all neurons active during transition (whose coincident activity with transition could be random) rather than those selectively active while BIL or BID (*Figure 2—figure*

*supplement 1M*). Thus, neurons which appear to best discriminate zone transition based on acute shifts in relative valence are also DP neurons. These data support a role for patchy neurons in locomotor restraint during LDb transitions.

We next analyzed GRIN-derived net fluorescence. A significant difference existed in patchy neuron activity surrounding transitions into light versus dark zones (GLME fixed effects 95% CI (a.u.) in brackets: ΔF/F is increased by BIL for approach [0.39086, 0.78166] and retreat [0.35765, 0.75611], *Figure 3D and E*). At face value, the pattern of greater average fluorescence corresponding to periods of greater expected speed (i.e. BIL) might be taken to reflect a positive correlation between calcium transients and locomotor output. However, speed itself failed to show the expected separation upstream of zone transitions for the imaged mice (*Figure 2—figure supplement 1H*, *Figure 2—figure supplement 1I*). Peak ΔF/F during transition into dark occurred while BIL, but did so early in the approach, below half-maximal speeds (*Figure 3D*). To test correlation to ΔF/F, acceleration events (defined as 4 s windows centered on acceleration moments and with mean speed exceeding 5 cm/s) were extracted across the entire LDbox session. R-values derived from events with significant (p<0.05) cross-correlation between ΔF/F and either speed or else acceleration were averaged for each mouse. Mean R-values representing the direction of correlation between ΔF/F and speed (*Figure 3F*) or else acceleration (*Figure 3G*) for each animal indicate a negative correlation to acceleration exclusively during transition into the light zone. Findings of greater BIL activity and of ΔF/F correlating with deceleration during transition into light, complement PA data implicating patch neurons in locomotor restraint during BIL transition moments.

## Patchy striatonigral neurons encode speed and deceleration

The patchy neurons we ablated and imaged within the dorsal striatum include both striatopallidal and striatonigral neurons, projecting to the *globus pallidus externus* (GPe), *globus pallidus internus* (GPi), and *substantia nigra pars reticulata* (SNr) as indicated by fluorescent markers tdTomato and SypGFP, a fusion of synaptophysin and green fluorescent protein (*Figure 4A*). Their efferents to these different target nuclei may convey distinct behavior-related information. During LDbox behavior, we used fiber photometry to collect the activity of patchy neuron axon terminals at each of these efferent nuclei (*Figure 4B*). We began by examining efferent activity at zone transitions and found that patchy neuronal activity clearly distinguished transitions into the light versus dark zone only at the level of the SNr (*Figure 4C*, *Figure 4—figure supplement 1A*). Whereas BIL moments surrounding zone transition increased net ΔF/F in the dorsal striatum, greater SNr efferent activity surrounding transitions into the light zone rendered the impact of BIL on ΔF/F pronouncedly negative on approach and positive on retreat at this nucleus (*Figure 4C*, *Figure 4—figure supplement 1B*). We proceeded to identify all super-threshold acceleration events as previously described to test correlation between ΔF/F and both speed and acceleration. Only SNr efferents displayed reliable correlation to both speed and to acceleration during area transitions (*Figure 4D*). GPe efferents showed small negative correlation to speed exclusively during transition into Light and GPi efferents showed a positive correlation to speed during transition into Light and stay time in Light, yet correlation lags were not significantly different from 0 (*Figure 4—figure supplement 1C–E*). No relationship to acceleration was identified for GPe, and a small negative correlation to acceleration during stay time in dark zone for GPi again showed non-directional lag (*Figure 4—figure supplement 1G–J*). These data implicate patchy neuron efferents synapsing in the SNr region in modulating LDb transition speeds.

Further examination of patchy neuron SNr efferent relationships to speed and acceleration demonstrated cross-correlation consistency across animals (*Figure 4E–H*). Positive correlation to speed and negative correlation to acceleration were confirmed across all conditions of zone transition and zone stay time (*Figure 4I–L*). Positive cross-correlational lags relating ΔF/F to speed-accompanied stay time in either zone, indicating that speed changed in advance of ΔF/F (*Figure 4J*). Negative lags relating ΔF/F to acceleration accompanied zone transitions and stay time in dark zone, indicating that ΔF/F changes preceded deceleration (*Figure 4L*). Lastly, we compared SNr efferent activity event frequency between zones and found fluorescent transients to be more frequent in the dark zone than in the light zone (p=0.0001, *Figure 4M*). This event frequency result was consistent across GPe and GPi efferents, and on re-examining patchy neuron activity collected through dorsal striatal GRIN lens (GPe p=0.0018, GPi p=0.0035, DS p=0.0373, *Figure 4—figure supplement 1F, L*). These data indicate that patchy striatonigral neurons encode speed and deceleration via efferents terminating

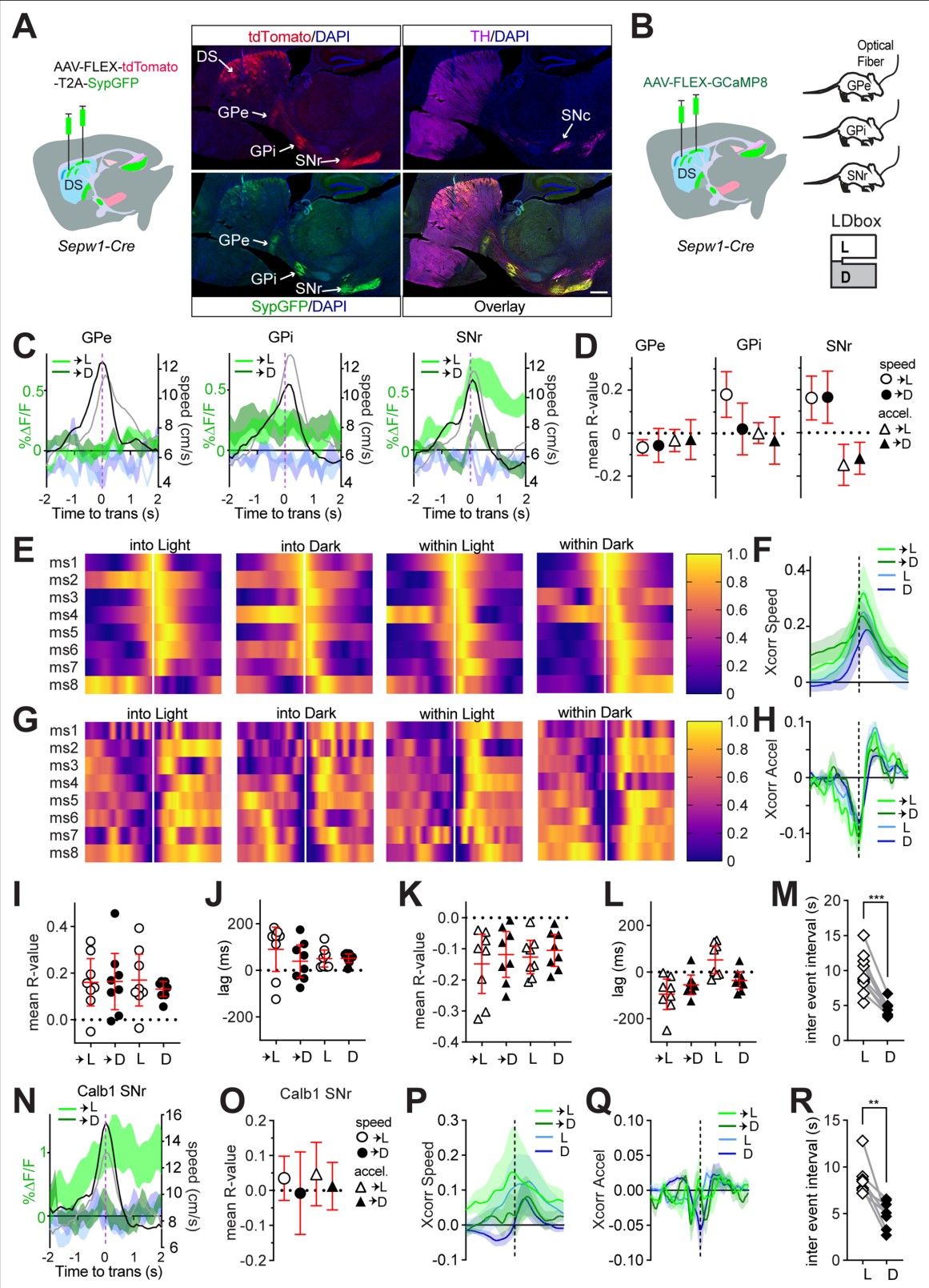

**Figure 4.** Patchy spiny projection neuron (SPN) efferents to *substantia nigra pars reticulata* (SNr) encode speed and deceleration. Sepw1-Cre: n=8 (4M,4F) GPe, n=9 (5M,4F) GPi, n=8 (4M,4F) SNr; *Calb1-Cre*: n=8 SNr. 95% CI shown in red. (**A**) Schematic of AAV1-phSyn1(S)-FLEX-tdTomato-T2A-SypEGFP injection into the dorsal striatum of a Sepw1-Cre mouse. A representative sagittal section shows patchy SPNs cell bodies and their efferents in the DS, GPe, GPi, and SNr from >3 mice. Scale bar 500 μm. (**B**) Injection schematic for mice with later fiber implants at GPe, GPi, or SNr. (**C**) Patchy

*Figure 4 continued on next page*

*Figure 4 continued*

SPN efferent activity aligned to zone transition, overlaid with speed. GCaMP8s into Light (light green) or Dark (dark green); simultaneous 405 nm channel into Light (light blue) or Dark (dark blue); mean speed (cm/s) during transition into Light (gray) or Dark (black). (**D**) Average across mice of mean R-value for acceleration events significantly correlated to ΔF/F (p<0.05) during zone transitions. (**E–H**) Patchy SPN ΔF/F at SNr cross-correlation to speed or acceleration aligned to acceleration events. Cross-correlogram color code: transition into Light (light green) or Dark (dark green), or within Light (light blue) or Dark (dark blue). (**E**) Heat map of cross-correlation to speed for each mouse. (**F**) ΔF/F cross-correlogram to speed. (**G**) Heat map of cross-correlation to acceleration for each mouse. (**H**) ΔF/F cross-correlogram to acceleration. (**I–L**) 95% CI in red. (**I, J**) For all acceleration events with significant (p<0.05) ΔF/F cross-correlation to speed, (**I**) mean R value per mouse, (**J**) mean Xcorr lag per mouse. (**K, L**) for all acceleration events with significant (p<0.05) ΔF/F cross-correlation to acceleration, (**K**) mean R value per mouse, (**L**) mean Xcorr lag per mouse. (**M**) Patchy SPN efferent ΔF/F at SNr inter-event interval in either zone, paired t-test, ***p=0.0001. (**N–Q**) Matrix ΔF/F at SNr (n=8; 4 M,4F). (**N**) Matrix efferent activity at SNr aligned to zone transition, overlaid with speed. Color coding identical to C. (**O**) Average across mice of mean R-value for acceleration events significantly correlated to ΔF/F (p<0.05) during zone transitions. (**P, Q**) Cross-correlogram color code identical to F and H. (**P**) ΔF/F cross-correlogram to speed. (**Q**) ΔF/F cross-correlogram to acceleration. (**R**) Matrix efferent ΔF/F at SNr inter-event interval in either zone, paired t-test, **p=0.0039.

The online version of this article includes the following figure supplement(s) for figure 4:

**Figure supplement 1.** Fiber photometry supplemental.

within SNr, where they may reach both SNr GABAergic and dopaminergic neurons (*Nadel et al., 2021*; *Dong et al., 2025*; *Okunomiya et al., 2025*). Moreover, these data show that activity of patchy striatonigral neurons precedes and may, therefore, lead to locomotor deceleration, and that patch neurons are more frequently active while mice reside within the dark zone of LDb.

We then investigated whether other types of striatal neurons, particularly those residing in the surrounding matrix compartments, play a similar role in LDb. We replicated the above photometry experiment and analyses in *Calbindin* 1 (*Calb1*)-*Cre* mice with fiber implants to SNr to record activity of matrix striatonigral neuron projections at this nucleus during LDb behavior (*Figure 4—figure supplement 1M*). Like patchy neuron efferents, matrix neuronal activity clearly distinguished transitions into light versus dark zone at the level of the SNr (*Figure 4N*) and demonstrated positive impact of BIL on ΔF/F during transition retreat (*Figure 4—figure supplement 1N*). However, matrix neuron efferent activity correlated to neither speed nor acceleration at zone transitions (*Figure 4O*). Examination of cross-correlation for all super-threshold acceleration events throughout the LDb revealed a modest positive correlation between ΔF/F and speed in only the dark zone with positive lag (*Figure 4P*, *Figure 4—figure supplement 1O, P*), whereas no correlation was identified between ΔF/F and acceleration (*Figure 4Q*, *Figure 4—figure supplement 1Q*). As with patchy neurons at all sampled locations, matrix neuronal activity at SNr also showed fluorescent transients to be more frequent in the dark zone than in the light zone (p=0.0039, *Figure 4R*). These results indicate that, despite similarity in coarse firing patterns at SNr, patchy neuron efferent activity is strongly tied to LDbox speed and deceleration at zone transitions, while matrix neuron efferent activity is not. Moreover, patchy neuron efferent activity at SNr may be causally related to deceleration due to negative cross-correlational lag, thereby affecting reductions in animal speed.

## Chemogenetic enhancement of patch neuronal activity increases rest and eliminates discriminative speed at choicepoints

Given that ablation of patch neurons enhanced valence-related speed and that our data from single-cell calcium imaging in the striatum and photometry at the SNr implicate patchy striatonigral neurons in locomotor deceleration, we sought to test for causality between patchy neuronal activity and LDb speed using chemogenetic manipulation with Designer Receptors Exclusively Activated by Designer Drugs (DREADD) in vivo. We hypothesized that acutely suppressing patchy neuronal activity by the Gi-coupled inhibitory DREADD (hM4Di) would elevate anxious vigor, matching observations in PA mice, and that acutely activating patch neuronal activity by the Gq-coupled excitatory DREADD (hM3Dq) would have the opposite effect – increasing restful exploration and lowering BIL speed at transitions. Using the same coordinates employed for ablation, we bilaterally infused *Sepw1-Cre* mice with one of three viruses: AAV-DIO-hM4Di-mCherry, AAV-DIO-hM3Dq-mCherry, or AAV-DIO-mCherry (control) (*Figure 5A*, *Figure 5—figure supplement 1*). Histology confirmed colocalization of mCherry-tagged DREADDs with patch marker MOR1 (*Figure 5B*). As with ablation, histological examination indicated a substantial fraction of dorsal patch territories identified through MOR1 staining were impacted (*Figure 5C*). Following 4–6 weeks' recovery, we tested mice on LDb, LLb, and DDb tests as

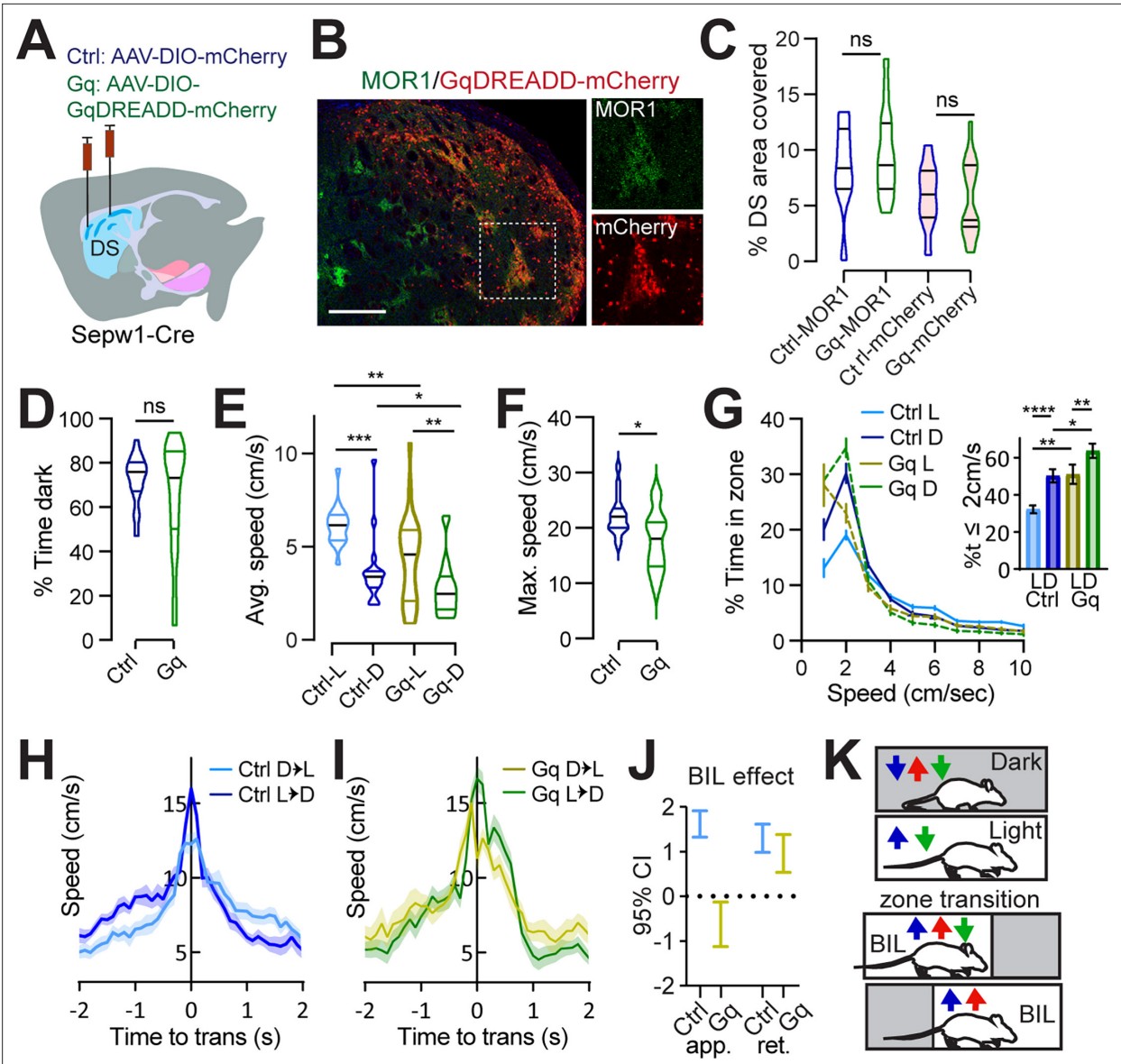

**Figure 5.** Patchy spiny projection neuron (SPN) enhancement increases rest and eliminates discriminative speed at choice points. (**A**) Injection schematic. (**B**) Dorsal striatal DREADD-mCherry overlaps patch marker MOR1 from >3 mice. Scale bar: 500 μm. 20× channel separation at right. (**C**) Dorsal striatal quantification of MOR1 and mCherry in Ctrl or Gq-DREADD-transduced mice. N=one dorsal striatal subregion (from Bregma in mm: R: 0.98–1.8, M: 0.5–0.7, C: 0.02–0.26) average value from n=4–6 sections per subregion, representing 3 Ctrl and 5 Gq mice. (**D–J**) Light/Dark box. n=17 (9M,8F) Ctrl and 20 (12M,8F) Gq mice. (**D**) % time in dark, p>0.05. (**E**) Average speed. Ctrl-L vs Ctrl-D, ***p=0.0001, Gq-L vs Gq-D, p=0.0037; Ctrl-vs-Gq in Light **p=0.0044, in Dark *p=0.0417. (**F**) Maximum speed. *p=0.01. (**G**) Speed distribution normalized to zone. **Insert**, % time ≤2 cm/s L vs D Ctrl ****p<0.0001, Gq **p=0.0056; Ctrl vs Gq in Light **p=0.0059, in Dark *p=0.0168. (**H–J**) Transition speed for controls (**H**) and Gq (**I**). (**J**) 95% C.I. for Generalized Linear Mixed-Effect (GLME) by group shows speed approaching transitions is increased by body-in-light (BIL) for controls but reduced for Gq-mice. (**K**) Cartoon summarizing speed modulation by valence under control (blue) SA (red) and Gq (green) conditions.

The online version of this article includes the following figure supplement(s) for figure 5:

**Figure supplement 1.** Chemogenetics supplemental.

before. Intraperitoneal injection of 0.3 mg/kg of DREADD-activating ligand JHU37160 (*Bonaventura et al., 2019*) preceded tests in all mice by 30–40 min.

All groups demonstrated unaltered, basic LDb performance in that stay-time was greater in the dark (*Figure 5D*) and speeds were higher in the light zone (control L vs D p=0.0001, Gq L vs D p=0.0037, *Figure 5E*; *Figure 5—figure supplement 1A, B*). The average speed within zone and maximum speed obtained were lower for Gq compared to control mice (average in L p=0.0044, in D

p=0.0417, *Figure 5E*, maximum p=0.01, *Figure 5F*), demonstrating mild locomotor suppression by Gq, and unchanged for Gi mice (*Figure 5—figure supplement 1B, C*). Overall, acute chemogenetic manipulation of patch neurons had little effect on classic measures of LDb performance. This corroborates findings from patch neuron ablation, reinforcing that patchy neuronal activity does not critically determine valence perception or grossly impact affect.

Relative to controls, Gq-mice spent more similar time at rest (≤2 cm/s) in either zone, with the most pronounced increase in percent time spent at rest compared to controls occurring within the light zone (L vs D control p<0.0001, Gq p=0.0056; control vs Gq L p=0.0059, D p=0.0168, *Figure 5G*). Meanwhile, Gi-mice showed no difference from controls in speed distribution (*Figure 5—figure supplement 1D*). Potential explanations for negative results from Gi-mice include already low basal activity among patchy neurons, or else the possibility that striatal neurons express insufficient GIRK channels for Gi-coupling to affect neuronal inhibition (*Shan et al., 2022*). In terms of transition speeds, control- and Gi-mice significantly discriminated alternative zone transitions as was seen in the ablation cohort - through enhanced vigor during BIL periods (*Figure 5H*, *Figure 5—figure supplement 1E*). In contrast, transition speed discrimination was impaired for Gq-mice through the apparent loss of BIL vigor (GLME fixed effects by group 95% CI (cm/s) in brackets: BIL increases approach speed for controls [1.3185, 1.9076] but decreases approach speed for Gq [–0.13168, –1.1211]. BIL increases retreat speed for controls [0.9823, 1.615] and Gq [0.53604, 1.3761], *Figure 5I and J*). These data confirm that enhanced patchy neuronal activity restricts LDb speed. Moreover, they illustrate that patchy neurons suppress anxiety-related vigor, increasing time spent at restful speeds in the light, and lowering BIL speed approaching transitions (*Figure 5K*).

Under homogeneous LL/DDbox conditions, no difference distinguished LLb from DDb performance for any group, and LLb and DDb performance was like controls for chemogenetically manipulated mice (*Figure 5—figure supplement 1F–M*). Whereas patch-enhanced Gq-mice selected lower LDb speeds in the context of a valence differential, there was essentially no difference between the performance of Gq-mice and controls in LLb and DDb (*Figure 5—figure supplement 1J–M*). Therefore, as with speed modulation in PA mice, speed modulation by DREADD-enhanced patchy neuronal activity was dependent on the presence of a valence differential.

## Patch/matrix balance at SNr controls Light/dark box locomotor phenotype

Having confirmed that broad enhancement of patchy neuronal activity led to valence-based slowing during LDb behavior, we sought to test the effect of targeted patchy neuron enhancement. Following photometry results implicating patchy striatonigral neuron terminals within SNr in LDb zone discrimination and deceleration, we chose to selectively enhance these terminals. Sepw1-Cre mice were bilaterally injected with one of two viruses: AAV-DIO-hChR2-mCherry (ChR mice) or AAV-DIO- mCherry (control) and were later bilaterally implanted with optical fibers above SNr (*Figure 6A and B*). LDb testing was carried out at least one week after implantation, and was run with continuous bilateral illumination (465 nm, 1 mW) for all mice throughout the 10 min on the maze.

Both groups demonstrated unaltered, basic LDb performance with greater stay time in the dark zone (*Figure 6C*) and greater speed in the light zone (control L vs D p<0.0001, ChR L vs D p=0.0011, *Figure 6D*). Overall, mice with channel rhodopsin at SNr performed similarly to controls. Neither average nor maximum speed distinguished ChR mice from controls, and speed histograms overlapped for the two groups (*Figure 6D–F*). A difference between groups was only observed in transition speeds when crossing between light and dark zones (*Figure 6G–J*). Whereas controls demonstrated the expected phenotype of enhanced BIL speed (i.e. while approaching dark zone and retreating into light zone, *Figure 6G*), this reflection of acute zone discrimination through speed was blunted in ChR mice (*Figure 6H*). Unlike controls, speed enhancement by BIL was quite small for ChR mice during transition approach (GLME fixed effects by group 95% CI (cm/s) in brackets: BIL increases approach speed for controls [1.041, 1.6462] more than for ChR [0.05499, 0.5939]. BIL increases retreat speed for controls [1.3526, 0.7299] and ChR [1.034, 0.5346], *Figure 6I*). This result matches findings from gross patchy neuronal activity enhancement using DREADDs, which also reduced BIL vigor approaching transition (*Figure 6J*). It is possible that a weaker phenotype was observed after optogenetically targeting SNr compared to chemogenetically targeting DS due to light scatter reaching fewer SNr fibers, or that additional neurons were involved in the broader locomotor suppression achieved with

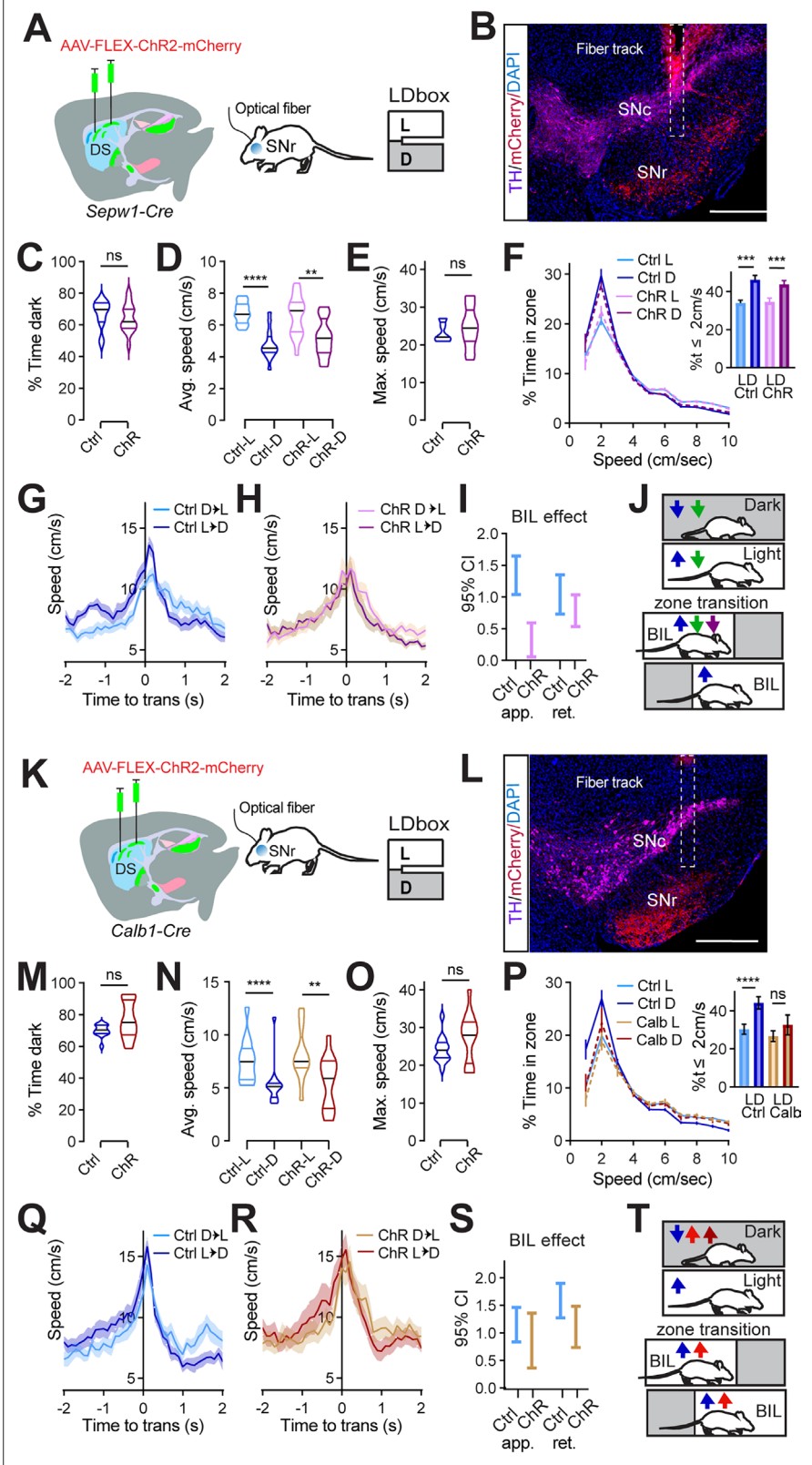

**Figure 6.** Patch/matrix balance at *substantia nigra pars reticulata* (SNr) controls Light/dark box locomotor phenotype. (**A**) Injection schematic. (**B**) Sample image of TH (magenta) and mCherry (red) staining in SN from >3 mice. The fiber track is indicated with dashed line. Scale bars: 500 μm. (**C–I**) N=14 (7M,7F) Ctrl and 16 (8M,8F) ChR Sepw1-Cre mice. (**C**) % time in dark, p>0.05. (**D**) Average speed. Ctrl-L vs Ctrl-D, ****p<0.0001, ChR-L

*Figure 6 continued on next page*

*Figure 6 continued*

vs ChR-D, **p=0.0011; Ctrl-vs-ChR in either zone p>0.05. (**E**) Maximum speed. p>0.05. (**F**) Speed distribution normalized to zone. **Insert**, %time ≤2 cm/s L vs D Ctrl ***p=0.0002, ChR ***p=0.0005; Ctrl vs ChR in either zone p>0.05. (**G–I**) Transition speed for controls (**G**) and ChR (**H**). (**I**) 95% CI coefficient for Generalized Linear Mixed-Effect (GLME) by group shows speed approaching transitions is increased by body-in-light (BIL) for controls significantly more than for ChR mice. (**J**) Cartoon summarizing speed modulation by valence under control (blue) Gq (green) and SNr ChR (purple) conditions for SpCre mice. (**K**) injection schematic. (**L**) Sample image of TH (magenta) and mCherry (red) staining in SN. The fiber track is indicated with dashed line. Scale bars: 500 µm. (**M–S**) n=13 (7M,6F) Ctrl and 13 (7M,6F) ChR *Calb1-Cre* mice. (**M**) % time in dark, p>0.05. (**N**) Average speed. Ctrl-L vs Ctrl-D, ****p<0.0001, ChR-L vs ChR-D, **p=0.0012; Ctrl-vs-ChR in either zone p>0.05. (**O**) Maximum speed. p>0.05. (**P**) Speed distribution normalized to zone. **Insert**, %time ≤2 cm/s L vs D Ctrl ****p<0.0001, ChR p>0.05; Ctrl vs ChR in either zone p>0.05. (**Q–S**) Transition speed for controls (**Q**) and ChR (**R**). (**S**) 95% CI coefficient for GLME by group. (**T**) Cartoon summarizing speed modulation by valence under control (blue) SA (red) and *Calb1-Cre* SNr ChR (maroon) conditions.

chemogenetics. Regardless, this data shows patchy neuron action at SNr is sufficient to selectively reduce valence-driven locomotor speed.

To test the behavioral impact of SNr efferents originating in the matrix, we carried out an identical optogenetic experiment in *Calb1-Cre* mice (***Figure 6K–T***). Again, mice with channel rhodopsin at SNr performed similarly to controls in many measures. Both groups demonstrated unaltered, basic LDb performance with greater stay time in the dark zone (***Figure 6M***) and greater speed in the light zone (control L vs D p<0.0001, ChR L vs D p=0.0012, ***Figure 6N***). No group difference existed in average or in maximum speed (***Figure 6N and O***). However, speed histograms distinguished ChR mice from controls. Compared to controls, ChR mice spent less time at rest in the dark zone (***Figure 6P***). Whereas controls spent significantly more time at speeds less than or equal to 2 cm/s in the dark zone than in the light zone, this difference disappeared for ChR mice (control L vs D p<0.0001, ChR p>0.05, ***Figure 6P***). Both controls and ChR mice demonstrated the expected phenotype of enhanced BIL speed during zone transitions (***Figure 6Q-S***). In summary, optogenetically activating matrix striato-nigral neuron terminals at SNr failed to reproduce the effect seen when activating patchy neuron terminals, illustrating their functional distinction at this synapse. Moreover, activating matrix terminals at SNr reproduced the effect on the speed histogram seen after patchy neuron ablation (***Figure 6T***). This suggests that a shift in relative strength between patch and matrix pathways modulate LDb speed, so that when patchy neuron efferents are outweighed by matrix efferents (as in PA as well as *Calb1*-Cre ChR at SNr), animals rest less in the safety of the dark. These data are complementary in ascribing a role to patchy striatonigral neurons acting at the SNr to decelerate mice with respect to a valence differential.

## Discussion

By focusing on free exploration of an innate valence differential, the present work reveals a previously undescribed, ethologically relevant role for patchy striatonigral neurons in governing implicit speed selection. These findings extend, complement, and challenge various prior data. Since movement velocity could be regulated through dopamine-dependent plasticity of striatal direct and indirect pathway strength to serve operant performance, this capacity for plasticity might function to trans-late implicit motivation into action vigor, or speed (***Yttri and Dudman, 2016***). Our work not only demonstrates that patchy striatonigral neurons possess the capacity for locomotor speed regulation, but our unique focus on untrained, free locomotion in the Light/dark box shows that this patchy neuron subpopulation is indeed naturally acting to translate implicit motivation (due to naturalistic contextual valence) into changes in locomotor speed. Another recent study parsed evaluative roles for patchy neuron subsets and noted speed changes during conditioned place preference, but only in the absence of any neuronal manipulation (***Xiao et al., 2020***). Considering the present findings, this speed difference almost certainly reflected implicit motivation, analogous to the LDb control phenotype, but the impact of patchy neurons on speed itself remained untested. It has been shown that broad striatal activity positively correlates with speed (***Cui et al., 2013***), yet here, we find genetically defined patchy neurons, comprising approximately 85% direct and 15% indirect pathway neurons (***Gerfen et al., 2013***; ***Smith et al., 2016***), suppress speed. This suppression of speed under naturalistic conditions

is a substantial departure from often-cited, correlative reports linking patchy neuron activation due to psychomotor stimulants with hyperkinetic states (*Canales and Graybiel, 2000*; *Saka et al., 2004*). Moreover, the present data implicate patchy striatonigral neurons in slowing locomotion, specifically by acting within the SNr to decelerate mice. This finding is consistent with recent results from independent lines of patchy neuron-specific mouse models (*Lazaridis et al., 2024*; *Dong et al., 2025*; *Okunomiya et al., 2025*), contradicting the traditional view that the direct-pathway striatonigral neurons exclusively promote locomotion.

Potential mechanisms by which striatal patchy neurons reduce locomotion involve the suppression of dopamine availability within the striatum. Dopamine, primarily supplied by neurons in the SNc and VTA, broadly facilitates locomotion (*Gerfen and Surmeier, 2011*; *Dudman and Krakauer, 2016*). Recent studies have shown that direct activation of patchy neurons leads to a reduction in striatal dopamine levels, accompanied by decreased walking speed (*Nadel et al., 2021*; *Dong et al., 2025*; *Okunomiya et al., 2025*). Patchy neuron projections terminate in structures known as 'dendron bouquets,' which enwrap SNc dendrites within the SNr and can pause tonic dopamine neuron firing (*Crittenden et al., 2016*; *Evans et al., 2020*). The present work highlights a role for patchy striatonigral inputs within the SN in decelerating movement, potentially through GABAergic dendron bouquets that limit dopamine release back to the striatum (*Dong et al., 2025*). Additionally, intrastriatal collaterals of patch spiny projection neurons (SPNs) have been shown to suppress dopamine release and associated synaptic plasticity via dynorphin-mediated activation of kappa opioid receptors on dopamine terminals (*Hawes et al., 2017*). This intrastriatal mechanism may further contribute to the reduction in striatal dopamine levels and the observed decrease in locomotor speed, representing a compelling avenue for future investigation.

We found that recorded differences in neural activity distribution while mice resided in the light depended on whether they were approaching the dark zone or retreating into the light zone. This, along with our LL/DDbox controls, powerfully illustrates that the Light/dark box models a dynamic response to perceived situational valence rather than to simple light level. Patchy neuron activity among cell bodies showed increased activity while the animal body resided in the light (BIL) to either side of a zone transition, but this was not true for patchy striatonigral efferents reaching to SNr. This suggests that distinct patchy neuronal subsets may be active depending not only on the current position of the mouse in light or dark zone, but on their current motivational state. Patchy neuron terminals in SNr showed BIL positively impacted ΔF/F entering light zone, but a negative BIL effect approaching transition, with a dip in activity in this moment. We believe this dip in deceleration-encoding activity helps to shape the control phenotype of greater BIL speed upstream of transition, particularly since BIL vigor on transition approach was dampened by exogenously enhancing patch neuron activity with chemogenetic or optogenetic manipulations. Although activity of matrix neuron efferents also differed in BIL impact surrounding transition, they showed no dip prior to the transition, and their activity was not correlated to speed or acceleration as was patch neurons'. It seems likely that a relative balance between patchy and matrix neuronal activity ultimately shapes locomotor speed, as is evidenced by the same speed histogram effect being achieved either by enhancing matrix or by ablating patchy neurons. Even without explicit patch neuron modulation, the BIL effect prior to the transition was lost in GRIN-implanted, mini scope-mounted mice. It is interesting to consider that excess handling and wearing of miniscopes may have led to natural patch neuron enhancement in this cohort, which is a documented response to chronic stress (*Friedman et al., 2017*). Future experiments studying patchy neuron's roles in valence-driven behavior will need to be designed cleverly to highlight distinctly valued moments while controlling for natural patchy neuron modulation by stress.

The potential conflict between findings of more abundant light-preferring neurons and more frequent neural activity in dark zone can be resolved with interesting implications. Determination of zone preference was made if a neuron expressed significantly higher net fluorescence while in one zone as opposed to the other. Net fluorescence takes area under the curve into consideration and is impacted by transient frequency, amplitude, and width. However, event frequency does not take into account transient amplitude or width. Moreover, due to the limitation of temporal dynamics of GCaMP fluorescence, transients with less than 1 s of separation between peaks could not be reliably distinguished, so that an aggregate of calcium influx cooccurring at greater than 1 Hz would be counted as a single event. Thus, our data suggest that many patchy neurons exhibit greater calcium influx while the animal resides within the light, potentially owing to summation of multiple large calcium entry

events. At the same time, patchy neuron transients are more frequent within the dorsal striatum, GPe, GPi, and SNr (and matrix transients within SNr) while animals reside in the dark zone. It is appealing to consider that greater frequency of deceleration-encoding activity not only accompanies but promotes slower locomotion or more frequent pausing in the dark zone. Likewise, it is intriguing to consider that greater net fluorescence among patchy neurons in the light zone may translate to burst firing shown to differentially engage SNr-area efferents (*Evans et al., 2020*), potentially promoting locomotion through dopaminergic rebound firing.

Judicious interpretation of the present data must consider the technical limitations of the various methods and circuit-level manipulations applied. Patchy neurons are distributed unevenly across the extensive structure of the striatum, and their targeted manipulation is constrained by viral spread in the dorsal striatum. Somatic calcium imaging using single-photon microscopy captures activity from only a subset of patchy neurons within a narrow focal plane beneath each implanted GRIN lens. Similarly, limitations in light diffusion from optical fibers may reduce the effective population of targeted fibers in both photometry and optogenetic experiments. For example, the more modest locomotor slowing observed with optogenetic activation of striatonigral fibers in the SNr compared to the stronger effects seen with Gq-DREADD activation across the dorsal striatum could reflect limited fiber optic coverage in the SNr. Alternatively, it may suggest that non-striatonigral mechanisms also contribute to generalized slowing. Our photometry data do not support a role for striatopallidal projections from patchy neurons in movement suppression. The potential contribution of intrastriatal mechanisms, discussed earlier, remains to be empirically tested. Although the behavioral assays used were naturalistic, many of the circuit-level interventions were not. Broad ablation or widespread activation of patchy neurons and their efferent projections represent non-physiological manipulations. Nonetheless, these perturbation results are interpreted alongside more naturalistic observations, such as in vivo imaging of patchy neuron soma and axon terminals, to form a coherent understanding of their functional role.

Together with previous studies of naturally occurring shifts in patchy neuron activation, these data illustrate ethologically relevant roles for a subgroup of genetically defined patchy neurons in behavior. Recent works have shown cognitive conflict and stress lead to patchy neuron disinhibition, and to explicit choices favoring greater risk-taking (*Friedman et al., 2015*; *Friedman et al., 2017*). Similarly, our ablation data demonstrate that an innate cost-benefit choice present in Light/Dark box navigation leads to the patchy neuron-mediated implicit 'choice' to reduce speed in riskier settings (BIL). Together with earlier findings by Friedman, our results suggest that patchy neurons act to limit animals' responsiveness to external stressors. These findings may also be unified if lower anxiety-related speed manifests lower anxiety, and, therefore, greater risk-taking and exploration. While dark preference was robust to patchy neuron manipulation, future experiments are warranted to probe the important relationship between anxious locomotor vigor and anxiety itself. The lowering of BIL speed may also indicate a change in allocation of energy or attention. Patchy neuronal activity has been shown to positively correlate with task engagement and perseveration (*Jenrette et al., 2019*; *Friedman et al., 2020*, *Nadel et al., 2020*; *Nadel et al., 2021*), implying support for habitual modes. Thus, by lowering BIL speed, patchy neurons may mediate a shift away from high alert and toward habit. We suspect that, by dampening response vigor to external valence, patchy neurons could conserve energy in safe (dark zone), familiar, or well-learned scenarios, alleviating cognitive demand at choice points and under duress (*Friedman et al., 2015*; *Beste et al., 2017a*; *Beste et al., 2017b*).

The present findings may also be applied to interpret or predict patchy neuron's contribution to disorders of attention, mood, and motor control, including Parkinson's disease. However, because of the heterogeneity of patchy neurons, it will be important to employ additional mouse lines while continuing to study patchy neuron's modulation by diverse afferents, including the prefrontal cortex and anxiety-related basal nucleus of the stria terminalis (*Smith et al., 2016*; *Friedman et al., 2017*), and to parse patchy neuron's impact through efferent pathways including striosome-dendron bouquets and the lateral habenula (*Hong et al., 2019*; *Evans et al., 2020*), in order to fully grasp patchy neuron's importance in health and disease.

## Materials and methods

### Mice

Animal work followed guidelines approved by the Institutional Animal Care and Use Committees (IACUC) of the National Institute on Aging (NIA), NIH. Mice came from one of two lines: Sepw1-Cre (NP67) (gifted from Dr. Chip Gerfen of NIMH) or *Calb1*-IRES2-Cre-D (Stain #: 028532, The Jackson Laboratory). Mice were male and female on a C57BL/6 background and were 3–5 months old at the time of behavioral testing, except for two females aged 6 and 8 months to meet size requirements (>25 g) for undergoing GRIN lens implantation. Mice were housed in a 12 hr light/dark cycle with ad libitum access to food and water. Behavioral tests were performed during the light cycle.

### Stereotaxic injections

Stereotaxic survival surgery was performed with aseptic technique. Anesthetized mice were head-mounted to stereotaxic frames and Bregma and Lambda leveled to within 0.05 mm. Mice received an infusion of AAVs ($5.85×10^{13}$ g.c./mL AAV1-FLEX-taCasp3 from Vigene, $2.42×10^{13}$ g.c./mL AAV5-EF1a-DIO-mCherry-WPRE from Vigene, $2.1×10^{13}$ g.c./mL AAV9-hSyn-DIO-hM4D(Gi)-mCherry from Addgene, $2.1×10^{13}$ g.c./mL AAV9-hSyn-DIO-hM3D(Gq)-mCherry from Addgene, $2.8×10^{12}$ g.c./mL AAV9-syn-FLEX-jGCaMP8s-WPRE from Addgene, $2.3×10^{13}$ g.c./mL AAV1-EF1a-double floxed-hChR2(H134R)-mCherry-WPRE-HGHpA from Addgene) through a vertically held syringe (2 µL Neuros Syringe, Hamilton). Infusion of these AAVs occurred at two points bilaterally for a total of four infusion sites per mouse with the following volumes, and at the following coordinates in mm relative to Bregma: 0.7–0.8 µL at AP 0.5, ML ± 2.2, DV –3; 0.5–0.6 µL at AP 1.5, ML ± 1.8, DV –3.5. Mice receiving GCaMP6s for calcium imaging ($1.3×10^{13}$ g.c./mL AAV8-CAG-FLEX-CGaMP6s-WPRE.25.641 from Vigene) were infused into the right hemisphere only through a 1700 series Gastight Hamilton syringe with 34 gauge needle and the tip entering the head angled 30 degrees toward rostral, delivering 0.5–0.6 µL to the following coordinates relative to Bregma: AP: −0.6 mm, M/L: +2 mm, D/V: −3 mm (D/V travel at 30 degrees from vertical, to avoid scar tissue in the future tract of the GRIN implant). All infusions were controlled by a motorized stereotaxic infusion pump (Stoelting) at a rate of 50–75 nL/min, with a 5 min wait between infusion completion and needle withdrawal. Scalp was closed with sutures and a small amount of Vetbond tissue adhesive (3 M) and coated in antibiotic ointment. Mice were given 5 mg/kg subcutaneous ketoprofen in lactated Ringer's solution immediately following surgery, and daily for two subsequent days. Alternatively, mice were administered slow-release meloxicam subcutaneously for pain.

### GRIN lens implant

10–14 days following GCaMP6s infusion, mice underwent a second aseptic, stereotaxic survival surgery as detailed in *Zhang et al., 2019* to implant a 1 mm diameter Gradient-Index (GRIN) lens to dorsal striatum. Briefly, this procedure involved the precise, robot-guided, vacuum excavation of a cylindrical tissue window centered at (relative to Bregma) A/P: +0.9 mm, M/L: +2 mm, and reaching –1.7 to –2 mm DV from dura. In later surgeries, implant depth was guided by epifluorescence signal detected during surgery using a stereo microscope fluorescence adapter (Night Sea). After lens placement, scalp was replaced by dental cement, extending up the sides of the GRIN lens to secure the implant, and mice were administered subcutaneous ketoprofen for three days as described above. Following 4 weeks recovery, non-surgical miniscope base-mounting was performed, permitting removable placement of the miniscope prior to experiments as detailed in *Barbera et al., 2016*.

### Fiber implant

10–14 days following viral infusion, mice underwent a second aseptic, stereotaxic survival surgery to place fiber optics unilaterally for photometry (GCaMP8s) or else bilaterally for optogenetics (hChR2). Target coordinates (relative to Bregma) were as follows: GPe A/P: –0.35 mm, M/L: +/-1.95 mm, and reaching –3.5 DV from dura; GPi A/P: –1.95 mm, M/L: +/-1.34 mm, and reaching –4.5 mm DV from dura; SNr A/P: –3.1 mm, M/L: +/-1.5 mm, and reaching –4.2 to –4.4 mm DV from dura. After fiber placement, a small area of scalp was replaced by dental cement to hold the implant in place. Mice were administered slow-release meloxicam subcutaneously for pain.

## Behavior

### Light/dark box

The light/dark box (LDb) apparatus comprised a plexiglass chamber 40 cm wide and 29 cm deep, partially bisected into a U-shaped floor space with 10 cm wide passage between light and dark zones (each 18 cm × 29 cm). The light zone had white walls and was sub-lit by white light; the dark had black walls and was sub-lit by red light. Translucent chamber flooring diffused light and eliminated a visual cliff. A lux meter held at the zone division and perpendicular to the floor read 500 lux toward light versus 40 lux toward dark. The test was conducted in an otherwise dark room, to which mice acclimated 30 min prior to testing. For testing, each mouse was gently placed into the light zone and video recorded aerially for 10 min. Ethanol and water were used to wipe down chamber surfaces between mice. TopScan 3.0 (CleverSys Inc) was used for animal tracking in videos. Mice were scored as being either in the light zone or else in the dark zone, with a zone transition scored at the time (t=0) when the tail base follows the nose from one zone to the other. Surrounding zone transition, body-in-light (BIL) moments were defined by the tail base residing in light preceding a transition into dark and following a transition into light; body-in-dark (BID) moments were defined by the tail base residing in dark preceding a transition into light and following a transition into dark. Speed was extracted from mouse center of mass. Maximum speed attained was defined for each mouse as the greatest speed recorded for ≥20 behavioral video frames (30 frames per second collection rate). Data exported from TopScan were organized in R and MATLAB and analyzed in Prism 10. A total of 8–15 mice per group were used to detect moderate-to-large effects. Mice were randomly assigned, and their prior treatments were blinded to the experimenters.

### LL/DD box

The same chambers used for LDb were used, with either entirely red lighting and black walls (Dark/Dark box), or entirely white sub-lighting and white walls (Light/Light box). Because chambers were constructed of slick, black plastic, wall color was reversibly modified by the application or removal of adhesive dry-erase laminate. The side the mice were initially placed into was deemed Side A and opposite side (analogous to LDb dark zone in terms of order of entry) was deemed Side B.

### Behavior with DREADD

Mice received an intraperitoneal injection of JHU37160 at 0.3 mg/kg body weight 30 min before placement on the maze. Otherwise, testing was identical. A total of 8–15 mice per group were used to detect moderate-to-large effects. Mice were randomly assigned, and their prior treatments were blinded to the experimenters.

### Behavior with miniscope

Mice were anesthetized briefly with 2–5% isoflurane vapor in an induction chamber so that miniscopes could be head-mounted, and focus adjusted on the mouse recovering. Mice were placed in a transfer cage within the testing room and allowed to acclimate 30 min prior to testing. Imaging experiments occurred in six 5 min sessions separated by 5 min rests in the home cage with LED powered off for cooling; the light and dark sides were switched between third and fourth sessions to control for spatial preference and to extend exploration. Otherwise, testing was completed identically to Light/Dark box described above. A total of 5–8 mice per group were used to detect moderate-to-large effects. Mice were randomly assigned, and their prior treatments were blinded to the experimenters.

### Behavior with fiber photometry or optogenetics

After acclimation to the test room, mice were scuffed briefly to attach unilateral (photometry) or bilateral (optogenetic) implanted fibers to fiber optic cables using ceramic cannula. They were then given 5 min rest before placement in the maze. In the case of optogenetic experiments, 1 mW 465 nm blue light was delivered continuously beginning several seconds before placement on the maze and ending with the test. A total of 5–8 mice per group were used to detect moderate-to-large effects. Mice were randomly assigned, and their prior treatments were blinded to the experimenters.

## Histology and light microscopy

Anesthetized mice were transcardially perfused with ice-cold phosphate-buffered saline (PBS) followed by 4% paraformaldehyde (PFA, Electron Microscopy Sciences). Brains were post-fixed in 4% PFA overnight and cryoprotected in 30% Sucrose (Sigma-Aldrich). 40 µm coronal or sagittal brain slices were sectioned using Leica cryostat CM3050S (Leica Biosystems) and stored at 4 °C in 0.03% Sodium Azide (Sigma-Aldrich) in PBS until use. Sections were rinsed in 0.3 M Glycine (Thermo Fisher Scientific) in PBS for 20 min at room temperature to quench autofluorescence, washed 3×5 min in PBS, and then incubated in blocking solution, rocking at 4 °C overnight. The same blocking solution was used to block and to prepare all antibody dilutions and consisted of 0.3% Triton X-100 (Sigma-Aldrich), 10% Donkey serum (Sigma-Aldrich), and 1% bovine serum albumen (Sigma-Aldrich) in PBS. Primary antibodies specific to TH (Rabbit 1:1000, Thermo Fisher Scientific, #P40101150; Chicken 1:500, Aves Labs, TYH), GFP (Chicken 1:500, Aves Labs, #GFP-1020), RFP (Mouse, Rockland Inc, #200301379) and MOR1 (Rabbit 1:3000, ImmunoStar, #24216) were freshly diluted and incubated rocking at 4 °C for two nights. Following 3×5 min rocking in PBS, Hoechst 33342 (1:10,000, Sigma-Aldrich) or DAPI (1:10,000, Thermo Fisher Scientific, #62248) was combined in blocking solution together with secondary antibodies Alexa Fluor (Thermo Fisher Scientific) 488- (anti-chicken, #A11039), 568- (anti-Rb, #A10042; anti-Ms, #A-21124), or 647- (anti-Rb, #A21244). Secondary antibodies were diluted 1:500. Tissue was incubated in secondary antibody for 1 hr at room temperature, or else rocking at 4 °C overnight. Fluorescence images were acquired using a laser scanning confocal microscopy LSM 780 (Zeiss). A 10× objective lens with 10% overlap in tiling was used to capture entire striatal hemispheres, while a 20× objective lens was used to capture higher magnification zones of interest.

## Calcium image processing

Calcium images were processed and analyzed using scripts in MATLAB (Mathworks). Calcium images were first stabilized using motion correction toolbox NoRMCorre (*Pnevmatikakis and Giovannucci, 2017*). Constrained non-negative matrix factorization-based calcium image processing toolbox CaImAn-MATLAB (*Pnevmatikakis et al., 2016*; *Giovannucci et al., 2019*) was used to extract calcium signals. Images of all recording sessions were concatenated and temporally sub-sampled to half for calcium signal extraction. The calcium trace ($\Delta F/F$) was set to zero if the value was below 3× baseline noise level, and $\Delta F/F$ was then normalized by $\Delta F/F/\max (\Delta F/F)$ for further data analysis. For each session, a global cell map was generated, including all the extracted neurons in that session, with a neuron denoted as active in one trial if it displayed calcium transient above 3× baseline noise level. To register the spatial footprints of neurons identified across different sessions, the displacement fields of the correlation image among sessions were first calculated using the MATLAB function (imregdemons) to estimate the shift due to the remounting of the miniscope and slow shift between the brain tissue and GRIN lens over time. The displacement fields were then applied to cell spatial footprints to align the rest of the sessions to the first session. Cell registration toolbox CellReg (*Sheintuch et al., 2017*) was then used for cell registration. Both distance model and spatial footprint model were used. Cosine similarity was used in our data. Pair-wise cell-cell distance and spatial footprint similarity was calculated from all recording sessions of all mice to decide the threshold of whether two cells in different sessions were the same or not. $P_{same}$ of 0.5 was used to choose the threshold. The cell with the highest spatial footprint similarity was chosen as the cell pair (same cell) if there were multiple cells that were above the threshold. Cell-to-cell mapping indices were then generated to indicate the relationship of the cell identity among sessions.

## Fiber photometry

A Tucker-Davis Technologies (TDT) system was used to stimulate and collect GCaMP transients emitted at 465 nm along with simultaneous, non-specific emissions at 405 nm. Raw data were imported to MATLAB for analysis. A double exponential was fit to the demodulated signals and then was subtracted from the signals to flatten signals prior to normalization.

## Quantification and statistical analysis

Behavior and histology data were analyzed in Prism 10 software (GraphPad). Data normality or non-normality was determined by the D'Agostino-Pearson omnibus test before selection of parametric or non-parametric tests. Unless otherwise stated in the text, comparison between two groups was done

through paired or unpaired two-tailed $t$-tests, Wilcoxon signed rank, or two-tailed Mann-Whitney (as appropriate). Comparison among more than two groups was done through repeated measures ANOVA or (where missing data precluded ANOVA) through Mixed-effects analysis prior to post hoc analysis by the above-mentioned tests. Comparison of the distribution of categorical variables among neurons was done through application of Chi-square or Fisher's exact test. Significance was determined at the $p < 0.05$ level, and exact p-values are given for significant tests unless $p < 0.0001$, in which case $p < 0.0001$ is reported for brevity. Speed or $\Delta F/F$ surrounding zone transitions is analyzed $-2$ s to $-0.6$ s (approach) or $+0.6$ s to $+2$ s (retreat) from transition; significant divergence in these values is determined by non-overlap in 95% confidence intervals from generalized linear mixed-effects (GLME) models generated by MATLAB function fitglme with fixed effects of time, BIL, experimental group where applicable, and random effect of subject. Figure error bars represent standard error of the mean, and violin plots illustrate median and interquartile range. All data points are biological replicates. Detailed statistical analyses are provided in *Supplementary file 1*.

Identification of light-dark zone-related cells: For each cell, calcium activities in light and dark zones ($\Delta F/F_{light}$ vs $\Delta F/F_{dark}$) were compared using the Mann-Whitney U test. A cell was defined as light zone cell if $\Delta F/F_{light}$ is significantly greater than $\Delta F/F_{dark}$ ($p < 0.05$ /n, where n is the number of identified neurons in each mouse), and similarly, a cell was defined as dark zone cell if $\Delta F/F_{dark}$ is significantly greater than $\Delta F/F_{light}$.

Identification of speed cells: For each cell, the Spearman correlation coefficient (SCC) between $\Delta F/F$ and locomotion speed was calculated. A cell was defined as linear+ (or linear-) cell if the SCC is significant ($p < 0.05$ /n, where n is the cell number for a given mouse; linear+: positive SCC and linear-: negative SCC). For cells lacking a significant SCC relationship, we fitted the activity speed relationship with quadratic curve ($ax^2 + bx + c$). A cell was defined as quadratic + (a>0) and quadratic- (a<0) if the goodness of fit is above 0.3.

Relationship between neural activity and acceleration: We first calculated the movement acceleration and binarize the acceleration by a threshold of 0.4 cm/s$^2$. The acceleration-deceleration preference was measured by receiver operating characteristic (ROC) analysis (*Britten et al., 1992*; *Li et al., 2017*). ROC curves were calculated by comparing the distribution of calcium activity within acceleration period versus the distribution of calcium activity within deceleration period. We then calculated the acceleration/deceleration preference by adjusted area under ROC curve (auROC) i.e., (auROC-0.5)×2.

Identification of transition-active cells: Those neurons active above a threshold of 0.0238 AU (determined by group activity) any time from 2 s preceding to 2 s following t=0 zone transition moments were defined as active during transitions.

Correlation to speed or acceleration: For photometry data (and aggregate fluorescence from single-cell imaging), acceleration events were identified as 4 s windows centered on an acceleration onset and having average speed exceeding 6 cm/s. Acceleration events were then separated into those for which the central acceleration moment falls in one of four groups: within light, within dark, or within a transition window in either direction. The MATLAB function xcorr was used to calculate R-values and lags for cross-correlation between $\Delta F/F$ and speed or else acceleration for each of these groups using all qualifying acceleration events.

Event frequency detection: For photometry data (and aggregate fluorescence from single-cell imaging), fluorescent events were detected as peaks with minimum distance of 1 s and exceeding 1.5 standard deviations from the mean of the flattened, normalized signal. Mean peak interval is displayed.

## Materials availability

This study did not generate new mouse models or reagents.

## Acknowledgements

This work was supported by the Intramural Research Program of the National Institute on Aging (ZIA AG000959) and the National Institute on Drug Abuse (ZIA DA000603), National Institutes of Health. We thank the National Institute of Mental Health (NIMH) rodent behavioral core for assisting in

behavioral tests with maze space, hardware, and software for animal tracking, and technical guidance as needed. We also thank Cai and Lin lab members for their various inputs and supports.

## Additional information

### Funding

| Funder | Grant reference number | Author |
|---|---|---|
| National Institute on Aging | ZIA AG000959 | Huaibin Cai |
| National Institute on Drug Abuse | ZIA DA000603 | Da-Ting Lin |

The funders had no role in study design, data collection and interpretation, or the decision to submit the work for publication.

### Author contributions

Sarah Hawes, Data curation, Formal analysis, Supervision, Investigation, Visualization, Methodology, Writing - original draft, Writing – review and editing; Bo Liang, Data curation, Formal analysis, Methodology, Writing – review and editing; Braden Oldham, Breanna T Sullivan, Bin Song, Data curation, Writing – review and editing; Lupeng Wang, Data curation, Methodology, Writing – review and editing; Lisa Chang, Methodology; Da-Ting Lin, Conceptualization, Resources, Supervision, Funding acquisition, Writing – review and editing; Huaibin Cai, Conceptualization, Supervision, Funding acquisition, Visualization, Writing - original draft, Project administration, Writing – review and editing

### Author ORCIDs

Braden Oldham ⓘ https://orcid.org/0000-0002-4249-5502
Da-Ting Lin ⓘ https://orcid.org/0000-0003-0538-7892
Huaibin Cai ⓘ https://orcid.org/0000-0002-8596-6108

### Ethics

Animal work followed Animal Study Protocol 464-LNG-2027 approved by Institutional Animal Care and Use Committees (IACUC) of the National Institute on Aging (NIA), NIH. All surgery was performed under sodium pentobarbital anesthesia, and every effort was made to minimize suffering.

Reviewer #1 (Public review): https://doi.org/10.7554/eLife.106403.4.sa1
Reviewer #2 (Public review): https://doi.org/10.7554/eLife.106403.4.sa2
Reviewer #3 (Public review): https://doi.org/10.7554/eLife.106403.4.sa3
Author response https://doi.org/10.7554/eLife.106403.4.sa4

## Additional files

### Supplementary files

MDAR checklist

Supplementary file 1. Statistics Table.

Source data 1. Original data for each graph.

### Data availability

All data generated or analysed during this study are included in the manuscript and supporting files. Raw calcium traces and MATLAB code for cell extraction are available in GitHub (copy archived at *Hawes et al., 2025*) and Zenodo.

The following dataset was generated:

| Author(s) | Year | Dataset title | Dataset URL | Database and Identifier |
|---|---|---|---|---|
| Liang B | 2025 | Striosome calcium imaging dataset | https://doi.org/ 10.5281/zenodo. 17227559 | Zenodo, 10.5281/ zenodo.17227559 |

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
