## [Editor Report · eLife Assessment]

The manuscript provides **important** findings on how striatal projection neurons regulate spontaneous locomotion speed in the context of implicit motivation and distinct contextual valence. The manuscript presented **convincing** supporting evidence for the findings. This work will be of broad interest to neuroscientists in the fields of basal ganglia, movement control, and cognition.

---

## [Referee Report · Reviewer #1 (Public review)]

Summary:

This fundamental work employed multidisciplinary approaches and conducted rigorous experiments to study how a specific subset of neurons in the dorsal striatum (i.e., "patchy" striatal neurons) modulates locomotion speed depending on the valence of naturalistic contexts.

Strengths:

The scientific findings are novel and original and significantly advance our understanding of how the striatal circuit regulates spontaneous movement in various contexts.

Weaknesses:

This is extensive research involving various circuit manipulation approaches. Some of these circuit manipulations are not physiological. This is discussed.

---

## [Referee Report · Reviewer #2 (Public review)]

Hawes et al. investigated the role of striatal neurons in the patch compartment of the dorsal striatum. Using Sepw1-Cre line, the authors combined a modified version of the light/dark transition box test that allows them to examine locomotor activity in different environmental valence with a variety of approaches, including cell-type-specific ablation, miniscope calcium imaging, fiber photometry, and opto-/chemogenetics. First, they found ablation of patchy striatal neurons resulted in an increase in movement vigor when mice stayed in a safe area or when they moved back from more anxiogenic to safe environments. The following miniscope imaging experiment revealed that a larger fraction of striatal patchy neurons was negatively correlated with movement speed, particularly in an anxiogenic area. Next, the authors investigated differential activity patterns of patchy neurons' axon terminals, focusing on those in GPe, GPi, and SNr, showing that the patchy axons in SNr reflect movement speed/vigor. Chemogenetic and optogenetic activation of these patchy striatal neurons suppressed the locomotor vigor, thus demonstrating their causal role in the modulation of locomotor vigor when exposed to valence differentials. Unlike the activation of striatal patches, such a suppressive effect on locomotion was absent when optogenetically activating matrix neurons by using the Calb1-Cre line, indicating distinctive roles in the control of locomotor vigor by striatal patch and matrix neurons. Together, they have concluded that nigrostriatal neurons within striatal patches negatively regulate movement vigor, dependent on behavioral contexts where motivational valence differs.

The strengths of this work include the use of multiple experimental approaches, including genetic/viral ablation of patch neurons, miniscope single-cell imaging, as well as projection-specific recording of axonal activity by fiber photometry, and causal manipulation of the neurons by chemogenetic and optogenetics. Although similar findings were reported previously, the authors' results will be of value owing to multiple levels of investigation. In my view, this study will add to the important literature by demonstrating how patch (striosomal) neurons in the striatum controls movement vigor.

---

## [Referee Report · Reviewer #3 (Public review)]

Hawes et al. combined behavioral, optical imaging, and activity manipulation techniques to investigate the role of striatal patch SPNs in locomotion regulation. Using Sepw1-Cre transgenic mice, they found that patch SPNs encode locomotion deceleration in a light-dark box procedure through optical imaging techniques. Moreover, genetic ablation of patch SPNs increased locomotion speed, while chemogenetic activation of these neurons decreased it. The authors concluded that a subtype of patch striatonigral neurons modulates locomotion speed based on external environmental cues.

In the revision, the authors have largely addressed my concerns with additional explanation and discussion, although some of the key experiments to strengthen the authors' claim by identifying the function of specific cell populations remain to be conducted due to technical challenges. Nevertheless, the current results remain valuable and interesting to a wide audience in the field.

---

## [Author Response]

The following is the authors’ response to the previous reviews

**Reviewer #1 (Public review):**
Summary:This fundamental work employed multidisciplinary approaches and conducted rigorous experiments to study how a specific subset of neurons in the dorsal striatum (i.e., "patchy" striatal neurons) modulates locomotion speed depending on the valence of the naturalistic context.Strengths:The scientific findings are novel and original and significantly advance our understanding of how the striatal circuit regulates spontaneous movement in various contexts. Response: We appreciate the reviewer’s positive evaluation.Weaknesses:This is extensive research involving various circuit manipulation approaches. Some of these circuit manipulations are not physiological. A balanced discussion of the technical strengths and limitations of the present work would be helpful and beneficial to the field. Minor issues in data presentation were also noted**.**

We have incorporated the recommended discussion of technical limitations and addressed the physiological plausibility of our manipulations on Page 33 of the revised Discussion section. Specifically, we wrote:

“Judicious interpretation of the present data must consider the technical limitations of the various methods and circuit-level manipulations applied. Patchy neurons are distributed unevenly across the extensive structure of the striatum, and their targeted manipulation is constrained by viral spread in the dorsal striatum. Somatic calcium imaging using single-photon microscopy captures activity from only a subset of patchy neurons within a narrow focal plane beneath each implanted GRIN lens. Similarly, limitations in light diffusion from optical fibers may reduce the effective population of targeted fibers in both photometry and optogenetic experiments. For example, the more modest locomotor slowing observed with optogenetic activation of striatonigral fibers in the SNr compared to the stronger effects seen with Gq-DREADD activation across the dorsal striatum could reflect limited fiber optic coverage in the SNr.Alternatively, it may suggest that non-striatonigral mechanisms also contribute to generalized slowing. Our photometry data do not support a role for striatopallidal projections from patchy neurons in movement suppression. The potential contribution of intrastriatal mechanisms, discussed earlier, remains to be empirically tested. Although the behavioral assays used were naturalistic, many of the circuit-level interventions were not. Broad ablation or widespread activation of patchy neurons and their efferent projections represent non-physiological manipulations. Nonetheless, these perturbation results are interpreted alongside more naturalistic observations, such as in vivo imaging of patchy neuron somata and axon terminals, to form a coherent understanding of their functional role”.

**Reviewer #2 (Public review):**
Hawes et al. investigated the role of striatal neurons in the patch compartment of the dorsal striatum. Using Sepw1-Cre line, the authors combined a modified version of the light/dark transition box test that allows them to examine locomotor activity in different environmental valence with a variety of approaches, including cell-type-specific ablation, miniscope calcium imaging, fiber photometry, and opto-/chemogenetics. First, they found ablation of patchy striatal neurons resulted in an increase in movement vigor when mice stayed in a safe area or when they moved back from more anxiogenic to safe environments. The following miniscope imaging experiment revealed that a larger fraction of striatal patchy neurons was negatively correlated with movement speed, particularly in an anxiogenic area. Next, the authors investigated differential activity patterns of patchy neurons' axon terminals, focusing on those in GPe, GPi, and SNr, showing that the patchy axons in SNr reflect movement speed/vigor. Chemogenetic and optogenetic activation of these patchy striatal neurons suppressed the locomotor vigor, thus demonstrating their causal role in the modulation of locomotor vigor when exposed to valence differentials. Unlike the activation of striatal patches, such a suppressive effect on locomotion was absent when optogenetically activating matrix neurons by using the Calb1-Cre line, indicating distinctive roles in the control of locomotor vigor by striatal patch and matrix neurons. Together, they have concluded that nigrostriatal neurons within striatal patches negatively regulate movement vigor, dependent on behavioral contexts where motivational valence differs.

We are grateful for the reviewer’s thorough summary of our main findings.

In my view, this study will add to the important literature by demonstrating how patch (striosomal) neurons in the striatum control movement vigor. This study has applied multiple approaches to investigate their functionality in locomotor behavior, and the obtained data largely support their conclusions. Nevertheless, I have some suggestions for improvements in the manuscript and figures regarding their data interpretation, accuracy, and efficacy of data presentation

We appreciate the reviewer’s overall positive assessment and have made substantial improvements to the revised manuscript in response to reviewers’ constructive suggestions.

(1) The authors found that the activation of the striatonigral pathway in the patch compartment suppresses locomotor speed, which contradicts with canonical roles of the direct pathway. It would be great if the authors could provide mechanistic explanations in the Discussion section. One possibility is that striatal D1R patch neurons directly inhibit dopaminergic cells that regulate movement vigor (Nadal et al., Sci. Rep., 2021; Okunomiya et al., J Neurosci., 2025). Providing plausible explanations will help readers infer possible physiological processes and give them ideas for future follow-up studies.

We have added the recommended data interpretation and future perspectives on Page 30 of the revised Discussion section. Specifically, we wrote:

“Potential mechanisms by which striatal patchy neurons reduce locomotion involve the supression of dopamine availability within the striatum. Dopamine, primarily supplied by neurons in the SNc and VTA,broadly facilitates locomotion (Gerfen and Surmeier 2011, Dudman and Krakauer 2016). Recent studies have shown that direct activation of patchy neurons leads to a reduction in striatal dopamine levels, accompanied by decreased walking speed (Nadel, Pawelko et al. 2021, Dong, Wang et al. 2025, Okunomiya, Watanabe et al. 2025). Patchy neuron projections terminate in structures known as “dendron bouquets”, which enwrap SNc dendrites within the SNr and can pause tonic dopamine neuron firing (Crittenden, Tillberg et al. 2016, Evans, Twedell et al. 2020). The present work highlights a role for patchy striatonigral inputs within the SN in decelerating movement, potentially through GABAergic dendron bouquets that limit dopamine release back to the striatum (Dong, Wang et al. 2025). Additionally, intrastriatal collaterals of patch spiny projection neurons (SPNs) have been shown to suppress dopamine release and associated synaptic plasticity via dynorphin-mediated activation of kappa opioid receptors on dopamine terminals (Hawes, Salinas et al. 2017). This intrastriatal mechanism may further contribute to the reduction in striatal dopamine levels and the observed decrease in locomotor speed, representing a compelling avenue for future investigation.”

(2) On page 14, Line 301, the authors stated that "Cre-dependent mCheery signals were colocalized with the patch marker (MOR1) in the dorsal striatum (Fig. 1B)". But I could not find any mCherry on that panel, so please modify it.

We have included representative images of mCherry and MOR1 staining in Supplementary Fig. S1 of the revised manuscript.

(3) From data shown in Figure 1, I've got the impression that mice ablated with striatal patch neurons were generally hyperactive, but this is probably not the case, as two separate experiments using LLbox and DDbox showed no difference in locomotor vigor between control and ablated mice. For the sake of better interpretation, it may be good to add a statement in Lines 365-366 that these experiments suggest the absence of hyperactive locomotion in general by ablating these specific neurons.

As suggested by the reviewer, we have added the following statement on Page 17 of the revised manuscript: “These data also indicate that PA elevates valence-specific speed without inducing general hyperactivity”.

(4) In Line 536, where Figure 5A was cited, the author mentioned that they used inhibitory DREADDs (AAV-DIO-hM4Di-mCherrry), but I could not find associated data on Figure 5. Please cite Figure S3, accordingly.

We have added the citation for the now Fig. S4 on Page 25 of the revised manuscript.

(5) Personally, the Figure panel labels of "Hi" and "ii" were confusing at first glance. It would be better to have alternatives.

As suggested by the reviewer, we have now labeled each figure panel with a distinct single alphabetical letter.

(6) There is a typo on Figure 4A: tdTomata → tdTomato

We have made the correction on the figure.

**Reviewer #3 (Public review):**
Hawes et al. combined behavioral, optical imaging, and activity manipulation techniques to investigate the role of striatal patch SPNs in locomotion regulation. Using Sepw1-Cre transgenic mice, they found that patch SPNs encode locomotion deceleration in a light-dark box procedure through optical imaging techniques. Moreover, genetic ablation of patch SPNs increased locomotion speed, while chemogenetic activation of these neurons decreased it. The authors concluded that a subtype of patch striatonigral neurons modulates locomotion speed based on external environmental cues. Below are some major concerns:The study concludes that patch striatonigral neurons regulate locomotion speed. However, unless I missed something, very little evidence is presented to support the idea that it is specifically striatonigral neurons, rather than striatopallidal neurons, that mediate these effects. In fact, the optogenetic experiments shown in Fig. 6 suggest otherwise. What about the behavioral effects of optogenetic stimulation of striatonigral versus striatopallidal neuron somas in Sepw1-Cre mice?

Our photometry data implicate striatonigral neurons in locomotor slowing, as evidenced by a negative cross-correlation with acceleration and a negative lag, indicating that their activity reliably precedes—and may therefore contribute to—deceleration. In contrast, photometry results from striatopallidal neurons showed no clear correlation with speed or acceleration.

Figure 6 demonstrates that optogenetic manipulation within the SNr of Sepw1-Cre^+^ striatonigral axons recapitulated context-dependent locomotor changes seen with Gq-DREADD activation of both striatonigral and striatopallidal Sepw1-Cre^+^ cells in the dorsal striatum but failed to produce the broader locomotor speed change observed when targeting all Sepw1-Cre^+^ cells in the dorsal striatum using either ablation or Gq-DREADD activation. The more subtle speed-restrictive phenotype resulting from ChR activation in the SNr could, as the reviewer suggests, implicate striatopallidal neurons in broad locomotor speed regulation. However, our photometry data indicate that this scenario is unlikely, as activity of striatopallidal Sepw1-Cre^+^ fibers is not correlated with locomotor speed. Another plausible explanation is that the optogenetic approach may have affected fewer striatonigral fibers, potentially due to the limited spatial spread of light from the optical fiber within the SNr. Broad locomotor speed change in LDbox might require the recruitment of a larger number of striatonigral fibers than we were able to manipulate with optogenetics. We have added discussion of these technical limitations to the revised manuscript. Additionally, we now discuss the possibility that intrastriatal collaterals may contribute to reduced local dopamine levels by releasing dynorphin, which acts on kappa opioid receptors located on dopamine fibers (Hawes, Salinas et al. 2017), thereby suppressing dopamine release.

The reviewer also suggests an interesting experiment involving optogenetic stimulation of striatonigral versus striatopallidal somata in Sepw1-Cre mice. While we agree that this approach would yield valuable insights, we have thus far been unable to achieve reliable results using retroviral vectors. Moreover, selectively targeting striatopallidal terminals optogenetically remains technically challenging, as striatonigral fibers also traverse the pallidum, and the broad anatomical distribution of the pallidum complicates precise targeting. This proposed work will need to be pursued in a future study, either with improved retrograde viral tools or the development of additional mouse lines that offer more selective access to these neuronal populations as we documented recently (Dong, Wang et al. 2025).

In the abstract, the authors state that patch SPNs control speed without affecting valence. This claim seems to lack sufficient data to support it. Additionally, speed, velocity, and acceleration are very distinct qualities. It is necessary to clarify precisely what patch neurons encode and control in the current study.

We believe the reviewer’s interpretation pertains to a statement in the Introduction rather than the Abstract: “Our findings reveal that patchy SPNs control the speed at which mice navigate the valence differential between high- and low-anxiety zones, without affecting valence perception itself.” Throughout our study, mice consistently preferred the dark zone in the Light/Dark box, indicating intact perception of the valence differential between illuminated areas. While our manipulations altered locomotor speed, they did not affect time spent in the dark zone, supporting the conclusion that valence perception remained unaltered. We appreciate the reviewer’s insight and agree it is an intriguing possibility that locomotor responses could, over time, influence internal states such as anxiety. We addressed this in the Discussion, noting that while dark preference was robust to our manipulations, future studies are warranted to explore the relationship between anxious locomotor vigor and anxiety itself. We report changes in scalar measures of animal speed across Light/Dark box conditions and under various experimental manipulations. Separately, we show that activity in both patchy neuron somata and striatonigral fibers is negatively correlated with acceleration—indicating a positive correlation with deceleration. Notably, the direction of the cross-correlational lag between striatonigral fiber activity and acceleration suggests that this activity precedes and may causally contribute to mouse deceleration, thereby influencing reductions in speed. To clarify this, we revised a sentence in the Results section:

“Moreover, patchy neuron efferent activity at the SNr may causally contribute to deceleration, asindicated by the negative cross-correlational lag, thereby reducing animal speed.”. We also updated the Discussion to read: “Together, these data specifically implicate patchy striatonigral neurons in slowing locomotion by acting within the SNr to drive deceleration.”

One of the major results relies on chemogenetic manipulation (Figure 5). It would be helpful to demonstrate through slice electrophysiology that hM3Dq and hM4Di indeed cause changes in the activity of dorsal striatal SPNs, as intended by the DREADD system. This would support both the positive (Gq) and negative (Gi) findings, where no effects on behavior were observed.

We were unable to perform this experiment; however, hM3Dq has previously been shown to be effective in striatal neurons (Alcacer, Andreoli et al. 2017). The lack of effect observed in GiDREADD mice serves as an unintended but valuable control, helping to rule out off-target effects of the DREADD agonist JHU37160 and thereby reinforcing the specificity of hM3Dq-mediated activation in our study. We have now included an important caveat regarding the Gi-DREADD results, acknowledging the possibility that they may not have worked effectively in our target cells:

“Potential explanations for the negative results in Gi-DREADD mice include inherently low basal activity among patchy neurons or insufficient expression of GIRK channels in striatal neurons, which may limit the effectiveness of Gicoupling in suppressing neuronal activity (Shan, Fang et al. 2022).”

Finally, could the behavioral effects observed in the current study, resulting from various manipulations of patch SPNs, be due to alterations in nigrostriatal dopamine release within the dorsal striatum?

We agree that this is an important potential implication of our work, especially given that we and others have shown that patchy striatonigral neurons provide strong inhibitory input to dopaminergic neurons involved in locomotor control (Nadel, Pawelko et al. 2021, Lazaridis, Crittenden et al. 2024, Dong, Wang et al. 2025, Okunomiya, Watanabe et al. 2025). Accordingly, we have expanded the discussion section to include potential mechanistic explanations that support and contextualize our main findings.

**Reviewer #1 (Recommendations for the authors):**
Here are some minor issues for the authors' reference:(1) This work supports the motor-suppressing effect of patchy SPNs, and >80% of them are direct pathway SPNs. This conclusion is not expected from the traditional basal ganglia direct/indirect pathway model. Most experiments were performed using nonphysiological approaches to suppress (i.e., ablation) or activate (i.e., continuous chemo-optogenetic stimulation). It remains uncertain if the reported observations are relevant to the normal biological function of patchy SPNs under physiological conditions. Particularly, under what circumstances an imbalanced patch/matrix activity may be induced, as proposed in the sections related to the data presented in Figure 6. A thorough discussion and clarification remain needed. Or it should be discussed as a limitation of the present work.

We have added discussion and clarification of physiological limitations in response to reviewer feedback. Additionally, we revised the opening sentence of an original paragraph in the discussion section to emphasize that it interprets our findings in the context of more physiological studies reporting natural shifts in patchy SPN activity due to cognitive conflict, stress, or training. The revised opening sentence now reads: “Together with previous studies of naturally occurring shifts in patchy neuron activation, these data illustrate ethologically relevant roles for a subgroup of genetically defined patchy neurons in behavior.”

(2) Lines 499-500: How striato-nigral cells encode speed and deceleration deserves a thorough discussion and clarification. These striatonigral cells can target both SNr GABAergic neurons and dendrites of the dopaminergic neurons. A discussion of microcircuits formed by the patchy SPNs axons in the SNr GABAergic and SNC DAergic neurons should be presented.

We have added this point at lines 499–500, including a reference to a relevant review of microcircuitry. Additionally, we expanded the discussion section to address microcircuit mechanisms that may underlie our main findings.

(3) Line 70: "BNST" should be spelled out at the first time it is mentioned.

This has been done.

(4) Line 133: only GCaMP6 was listed in the method, but GCaMP8 was also used (Figure 4). Clarification or details are needed.

Thank you for your careful attention to detail. We have corrected the typographical errors in the Methods section. Specifically, in the Stereotaxic Injections section, we corrected “GCaMP83” to “GCaMP8s.” In the Fiber Implant section, we removed the incorrect reference to “GCaMP6s” and clarified that GCaMP8s was used for photometry, and hChR2 was used for optogenetics.

(5) Line 183: Can the authors describe more precisely what "a moment" means in terms of seconds or minutes?

This has been done.

(6) Line 288: typo: missing / in ΔF

Thank you this has been fixed

(7) Line 301-302: the statement of "mCherry and MOR1 colocalization" does not match the images in Figure 1B.

This has been corrected by proving a new Supplementary Figure S1.

(8) Related to the statement between Lines 303-304: Figure 1c data may reflect changes in MOR1 protein or cell loss. Quantification of NeuN+ neurons within the MOR1 area would strengthen the conclusion of 60% of patchy cell loss in Figure 1C

Since the efficacy of AAV-FLEX-taCasp3 in cell ablation has been well established in our previous publications and those of others (Yang, Chiang et al. 2013, Wu, Kung et al. 2019), we do not believe the observed loss of MOR1 staining in Fig. 1C merely reflects reduced MOR1 expression. Moreover, a general neuronal marker such as NeuN may not reliably detect the specific loss of patchy neurons in our ablation model, given the technical limitations of conventional cell-counting methods like MBF’s StereoInvestigator, which typically exhibit a variability margin of 15–20%.

(9) Lines 313-314: "Similarly, PA mice demonstrated greater stay-time in the dark zone (Figure 1E)." Revision is needed to better reflect what is shown in Figure 1E and avoid misunderstandings.

Thank you this has been addressed.

(10) The color code in Figure 2Gi seems inconsistent with the others? Clarifications are needed

Color coding in Figure 2Gi differs from that in 2Eii out of necessity. For example, the "Light" cells depicted in light blue in 2Eii are represented by both light gray and light red dots in 2Gi. Importantly, Figure 2G does not encode specific speed relationships; instead, any association with speed is indicated by a red hue.

(11) Lines 538-539: the statement of "Over half of the patch was covered" was not supported by Figure 5C. Clarification is needed.

Thank you. For clarity, we updated the x-axis labels in Figures 1C and 5C from “% area covered” to “% DS area covered,” and defined “DS” as “dorsal striatal” in the corresponding figure legends. Additionally, we revised the sentence in question to read: “As with ablation, histological examination indicated that a substantial fraction of dorsal patch territories, identified through MOR1 staining, were impacted (Fig. 5C).”

(12) Figure 3: statistical significance in Figure 3 should be labeled in various panels.

We believe the reviewer's concern pertains to the scatter plot in panel F—specifically, whether the data points are significantly different from zero. In panel 3F, the 95% confidence interval clearly overlaps with zero, indicating that the results are not statistically significant.

(13) Figures 6D-E: no difference in the speed of control mice and ChR2 mice under continuous optical stimulation was not expected. It was different from Gq-DRADDS study in Figure 5E-F. Clarifications are needed.

For mice undergoing constant ChR2 activation of Sepw1-Cre+ SNr efferents, overall locomotor speed does not differ from controls. However, the BIL (bright-to-illuminated) effect on zone transitions isdisrupted: activating Sepw1-Cre^+^ fibers in the SNr blunts the typical increase in speed observed when mice flee from the light zone toward the dark zone. This impaired BIL-related speed increase upon exiting the light was similarly observed in the Gq-DREADD cohort. The reviewer is correct that this optogenetic manipulation within the SNr did not produce the more generalized speed reductions seen with broader Gq-DREADD activation of all Sepw1-Cre^+^ cells in the dorsal striatum. A likely explanation is the difference in targeting—ChR2 specifically activates SNr-bound terminals, whereas Gq-DREADD broadly activates entire Sepw1-Cre^+^ cells. Notably, many of the generalized speed profile changes observed with chemogenetic activation are opposite to those resulting from broad ablation of Sepw1-Cre^+^ cells. The more subtle speed-restrictive phenotype observed with ChR2 activation targeted to the SNr may suggest that fewer striatonigral fibers were affected by this technique, possibly due to the limited spread of light from the fiber optic. Broad locomotor speed change in LDbox might require the recruitment of a larger number of striatonigral fibers than we were able to manipulate with an optogenetic approach. Alternatively, it could indicate that non-striatonigral Sepw1-Cre^+^ projections—such as striatopallidal or intrastriatal pathways—play a role in more generalized slowing. If striatopallidal fibers contributed to locomotor slowing, we would expect to see non-zero cross-correlations between neural activity and speed or acceleration, along with negative lag indicating that neural activity precedes the behavioral change. However, our fiber photometry data do not support such a role for Sepw1-Cre^+^ striatopallidal fibers. We have also referenced the possibility that intrastriatal collaterals could suppress striatal dopamine levels, potentially explaining the stronger slowing phenotype observed when the entire striatal population is affected, as opposed to selectively targeting striatonigral terminals. These technical considerations and interpretive nuances have been incorporated and clarified in the revised discussion section.

(14) Lines 632: "compliment": a typo?

Yes, it should be “complement”.

(15) Figure 4 legend: descriptions of panels A and B were swapped

Thank you. This has been corrected.

(16) Friedman (2020) was listed twice in the bibliography (Lines 920-929).

Thank you. This has been corrected.

**Reviewer #3 (Recommendations for the authors):**
It will be helpful to label and add figure legends below each figure.

Thank you for the suggestion.

Editor's note:Should you choose to revise your manuscript, if you have not already done so, please include full statistical reporting including exact p-values wherever possible alongside the summary statistics (test statistic and df) and, where appropriate, 95% confidence intervals. These should be reported for all key questions and not only when the p-value is less than 0.05 in the main manuscript. We noted some instances where only p values are reported.Readers would also benefit from coding individual data points by sex and noting N/sex

We have included detailed statistical information in the revised manuscript. Both male and female mice were used in all experiments in approximately equal numbers. Since no sex-related differences were observed, we did not report the number of animals by sex.

References

Alcacer, C., L. Andreoli, I. Sebastianutto, J. Jakobsson, T. Fieblinger and M. A. Cenci (2017). "Chemogenetic stimulation of striatal projection neurons modulates responses to Parkinson's disease therapy." J Clin Invest 127(2): 720-734.

Crittenden, J. R., P. W. Tillberg, M. H. Riad, Y. Shima, C. R. Gerfen, J. Curry, D. E. Housman, S. B. Nelson, E. S. Boyden and A. M. Graybiel (2016). "Striosome-dendron bouquets highlight a unique striatonigral circuit targeting dopamine-containing neurons." Proc Natl Acad Sci U S A 113(40): 1131811323.

Dong, J., L. Wang, B. T. Sullivan, L. Sun, V. M. Martinez Smith, L. Chang, J. Ding, W. Le, C. R. Gerfen and H. Cai (2025). "Molecularly distinct striatonigral neuron subtypes differentially regulate locomotion." Nat Commun 16(1): 2710.

Dudman, J. T. and J. W. Krakauer (2016). "The basal ganglia: from motor commands to the control of vigor." Curr Opin Neurobiol 37: 158-166.

Evans, R. C., E. L. Twedell, M. Zhu, J. Ascencio, R. Zhang and Z. M. Khaliq (2020). "Functional Dissection of Basal Ganglia Inhibitory Inputs onto Substantia Nigra Dopaminergic Neurons." Cell Rep 32(11): 108156.

Gerfen, C. R. and D. J. Surmeier (2011). "Modulation of striatal projection systems by dopamine." Annual review of neuroscience 34: 441-466.

Hawes, S. L., A. G. Salinas, D. M. Lovinger and K. T. Blackwell (2017). "Long-term plasticity of corticostriatal synapses is modulated by pathway-specific co-release of opioids through kappa-opioid receptors." J Physiol 595(16): 5637-5652.

Lazaridis, I., J. R. Crittenden, G. Ahn, K. Hirokane, T. Yoshida, A. Mahar, V. Skara, K. Meletis, K.Parvataneni, J. T. Ting, E. Hueske, A. Matsushima and A. M. Graybiel (2024). "Striosomes Target Nigral Dopamine-Containing Neurons via Direct-D1 and Indirect-D2 Pathways Paralleling Classic DirectIndirect Basal Ganglia Systems." bioRxiv.

Nadel, J. A., S. S. Pawelko, J. R. Scott, R. McLaughlin, M. Fox, M. Ghanem, R. van der Merwe, N. G. Hollon, E. S. Ramsson and C. D. Howard (2021). "Optogenetic stimulation of striatal patches modifies habit formation and inhibits dopamine release." Sci Rep 11(1): 19847.

Okunomiya, T., D. Watanabe, H. Banno, T. Kondo, K. Imamura, R. Takahashi and H. Inoue (2025).

"Striosome Circuitry Stimulation Inhibits Striatal Dopamine Release and Locomotion." J Neurosci 45(4).

Shan, Q., Q. Fang and Y. Tian (2022). "Evidence that GIRK Channels Mediate the DREADD-hM4Di Receptor Activation-Induced Reduction in Membrane Excitability of Striatal Medium Spiny Neurons." ACS Chem Neurosci 13(14): 2084-2091.

Wu, J., J. Kung, J. Dong, L. Chang, C. Xie, A. Habib, S. Hawes, N. Yang, V. Chen, Z. Liu, R. Evans, B. Liang, L. Sun, J. Ding, J. Yu, S. Saez-Atienzar, B. Tang, Z. Khaliq, D. T. Lin, W. Le and H. Cai (2019). "Distinct Connectivity and Functionality of Aldehyde Dehydrogenase 1a1-Positive Nigrostriatal Dopaminergic Neurons in Motor Learning." Cell Rep 28(5): 1167-1181 e1167.

Wu, J., J. Kung, J. Dong, L. Chang, C. Xie, A. Habib, S. Hawes, N. Yang, V. Chen, Z. Liu, R. Evans, B. Liang, L. Sun, J. Ding, J. Yu, S. Saez-Atienzar, B. Tang, Z. Khaliq, D. T. Lin, W. Le and H. Cai (2019). "Distinct Connectivity and Functionality of Aldehyde Dehydrogenase 1a1-Positive Nigrostriatal Dopaminergic Neurons in Motor Learning." Cell Rep 28(5): 1167-1181 e1167.